## Research Article

perinatal; mental health; maternal; mental illness

**Corresponding author:**
Nicole Votruba;
Email: Nicole.votruba@wrh.ox.ac.uk

# Perinatal mental health in India in the states of Haryana and Telangana: A district-level situational analysis

Lucy Mellers[1] , Sudhir Raj Thout[2,6] , Sandhya Kanaka Yatirajula[2] , Jane Hirst[3] , Devarsetty Praveen[2,4,7], Pallab K Maulik[2,4,7] , Vinod Bobji[5] and Nicole Votruba[1,3]

[1]Nuffield Department of Women's and Reproductive Health, University of Oxford, Oxford, UK; [2]The George Institute for Global Health India, New Delhi, India; [3]The George Institute for Global Health, Imperial College London, London, UK; [4]The George Institute for Global Health, University of New South Wales, Sydney, NSW, Australia; [5]District Medical and Health Office, Siddipet, Telangana, India; [6]Indian Institute of Technology, Hyderabad, India and [7]Prasanna School of Public Health, Manipal Academy of Higher Education, India

## Abstract

Women in the perinatal phase are at an increased risk of experiencing mental health problems, but in low and middle-income countries such as India, perinatal mental health (PMH) care provision is often scarce. This situational analysis presents the formative findings of the SMARThealth Pregnancy and Mental Health (PRAMH) project (Votruba et al. 2023). It investigates the nature and availability of maternal mental health policies, legislation, systems and services, as well as relevant context and community in India on a national, state (Haryana and Telangana) and district (Faridabad and Siddipet) level. A desktop, scoping review and informal interviews with mental health experts were conducted. Socio-demographic and maternal health indicators vary between Haryana and Telangana. No specific national PMH policy or plan is available. General mental health services exist at a district level within Siddipet and Faridabad, but no specific PMH services have been identified.

## Impact statement

In India, many women experience mental health conditions during the perinatal period, yet most of them do not receive treatment. The substantial negative impact of untreated perinatal mental health (PMH) conditions demands urgent, high-quality research into strategies to close this treatment gap. Interventions must be evidence-based, culturally tailored and consider the wider policy and health systems within which they sit. This situational analysis took a broad approach to understanding maternal health and socio-economic context as well as the systems and policies on a national, state and district level in two states, Haryana and Telangana. Using a mixed methods approach, we have mapped out the existing evidence on systems and policies, as well as delving into community-level barriers to PMH support. There are important socio-demographic differences between the two states, such as rates of adolescent pregnancy and intimate partner violence, as well as differences in maternal health outcomes and healthcare interactions. Understanding these factors will help shape future interventions as well as support policies for maternal mental health in India. Key barriers to seeking PMH support include stigma, isolation and transport difficulties. Calls are growing for the integration of mental health into routine maternal care. Delivering national and state-level plans and policies that are evidence-based and community-driven is essential. This situational analysis is part of the formative work of the Pregnancy and Mental Health project (PRAMH and provides an essential step for the co-development of a community-based intervention for women in the perinatal phase in Haryana and Telangana.

## Introduction

The perinatal period, which encompasses pregnancy and the first year after giving birth, is associated with a high risk of mental disorders, particularly depression and anxiety (Biaggi et al., 2016). Globally, up to 20% of women experience perinatal mental health conditions (PMHCs), with a higher prevalence in low- and middle-income countries (LMICs) (Fellmeth et al., 2021) where maternal mental illness is both under-recognised and under-treated (Gelaye et al., 2016), meaning the true disease burden is likely to be underestimated. Maternal mental health problems can have far-reaching consequences for both mothers and children. PMHCs are associated with

difficulties in maternal self-care and infant bonding (Śliwerski et al., 2020), leading to intergenerational impacts. The total costs for a hypothetical cohort of women with perinatal depression and anxiety, as well as their children living in Pakistan, were estimated to be $16.5 billion (Bauer et al., 2024).

PMHCs can affect the infant before birth, after birth and long into a child's development. Maternal depression is linked to stillbirth, premature birth and a reduction in birth weight (Räisänen et al., 2014). There are sequelae for the social and emotional development of the child: for instance, children of mothers with perinatal depression are more likely to show symptoms of attention deficit disorder and conduct disorder (Glover et al., 2014). In addition, postnatal depression is the strongest predictor of parenting stress and difficulties in the mother–infant relationship (Leigh and Milgrom, 2008), and is associated with an increased risk of depression in adolescence (Murray et al., 2011; Pearson et al., 2013).

Although maternal mortality in India has reduced by over 50% since the early 2000s (Singh, 2018) to 97 deaths per 100,000, maternal suicide constitutes an increasing proportion of maternal deaths. A recent report in Kerala estimated that maternal suicide accounted for nearly one in five maternal deaths in 2020 (Paily et al., 2020). Further progress towards the United Nations Sustainable Development Goal (UN SDG) for maternal mortality in India, therefore, depends on the urgent provision of better prevention and treatment for PMHCs in women.

Reducing the burden of PMHCs in India has been defined as a priority for research and clinical practice (Ganjekar et al., 2020; Desai and Chandra, 2023). India is the world's most populous country with several languages as well as cultural, religious and socio-economic diversity, and a largely rural population distribution. Services for maternal and mental health vary greatly across and within the country's 29 states (Ganjekar et al., 2020). In India, as in other LMICs, risk factors for mental disorders are more common (Varma et al., 2007; Howard et al., 2013; Choi et al., 2019) than in high-income countries (HICs) and include: poverty, lack of employment, physical health problems, issues of safety and security and adolescent pregnancy (Agnafors et al., 2019). Importantly, intimate partner violence has been identified as one of the strongest predictors for PMHCs (Howard et al., 2013; Halim et al., 2018).

Widespread availability of perinatal mental health (PMH) services is critical for reducing maternal morbidity and mortality, yet maternal mental health has not been a policy priority in India in the past decades (Ganjekar et al., 2020). Increasingly, in parts of the country, efforts have been made to call attention to maternal mental health (The New Indian Express, 2019; Fuhr et al., 2019; Ganjekar and Parthasarathy, 2023; UNICEF, 2023) such as the 'Amma Manasu' PMH programme in Kerala (Thiruvananthapuram, 2019) and the Thayi card in Karnataka (Ganjekar and Parthasarathy, 2023), which integrates mental health into routine antenatal care, a strategy recommended by World Health Organisation (WHO, 2022). Routine screening for PMHCs is associated with improved identification of women at risk, as well as increased referral and engagement with mental health services (Reilly et al., 2020) and is recommended in many countries including Australia (Highet and Purtell, 2012), the United States (O'Connor et al., 2016) and the United Kingdom (National Collaborating Centre for Mental Health, 2007). However, screening and treatment require an understanding of the specific challenges and socio-cultural context of PMH and the local setting (Shrestha et al., 2016). As India currently lacks a national maternal mental health programme, understanding existing efforts as well as challenges on a national, state and district level is important in shaping future policy.

This study presents the situational analysis for the formative phase of the SMARThealth PRegnancy And Mental Health (PRAMH) project (Votruba et al., 2023). The situational analysis will investigate the nature and availability of maternal mental health policies, legislation, systems and services, as well as relevant context and community in India on a national, state and district level. Conducting a situational analysis is a critical first step when planning for research and implementation of a study in contexts where mental health services and systems are under-resourced or lacking (Murphy et al., 2019). This presents the first step in co-developing an intervention to support women's PMH in Telangana and Haryana (Votruba et al., 2023). These two states are located in South and North India (Supplementary Materials Figure 1). Both states differ socio-culturally and in terms of health services. Telangana has maintained an average of less than 70 maternal deaths per 100,000 live births since 2017, and the most recent data makes it the third best performing state in the country. In comparison, in Haryana, maternal mortality rates are not only higher but have increased from 98 to 110 (Sample Registration System, 2022).

## Methods

This situational analysis focuses on the two sites of the PRAMH study: Siddipet District (Telangana state) and Faridabad District (Haryana state). We used a standardised tool (Hanlon et al., 2014) developed by the PRIME consortium to collect data for the planning of integrated mental healthcare. This tool focuses on factors required for the implementation of WHO's mhGAP (Dua et al., 2011) intervention guide. It was previously adapted for a maternal mental health context (Baron et al., 2016) and further modified for this study. The sections of the situational analysis were structured as follows:

1. Socio-economic and maternal health context,
2. Perinatal mental health policies and legislation,
3. Perinatal mental health services – mental health treatment coverage, district level health services,
4. Perinatal mental health and the community, and
5. Monitoring and evaluation.

### Data collection

Methods used to inform each section of the situational analysis are illustrated in Table 1. Secondary data was collected between February 2021 and April 2024 from publicly available sources. The scoping literature review was conducted between September 2022 and March 2025.

Major socio-demographic determinants were explored, such as access to sanitation, clean water and power, as well as female literacy, adolescent pregnancy and intimate partner violence.

Maternal health was investigated using the National Family Health Surveys and Sample Registration System Bulletins to describe key indicators of care access and potentially identify opportunities for future perinatal mental health integration. We considered key indicators of maternal health access, such as the availability of institutional or otherwise assisted deliveries and receipt of antenatal care.

In addition, informal interviews were conducted between June 2022 and November 2023 with a number of experts (psychiatrists, psychologists, psychiatric nurses), to inform sections 3 (perinatal mental health services) and 4 (community) of the analysis. Informal interviews were employed as they allow ease of conversation and

**Table 1.** Methods and data sources that have informed the situational analysis and interviews

| Situational analysis section | Methods used | Data sources |
|---|---|---|
| Socio-economic and maternal health context | Desktop review of relevant sites – collecting secondary data Google search for relevant statistics | Census of India https://www.censusindia.gov.in HMIS https://hmis.mohfw.gov.in National Family Health Survey https://rchiips.org/nfhs/factsheet_NFHS–5.shtml Telangana state website https://www.telangana.gov.in Crime in India https://ncrb.gov.in/uploads/nationalcrimerecordsbureau/custom/1701607577CrimeinIndia2022Book1.pdf Health Department Haryana https://haryanahealth.gov.in |
| Policies and legislation around perinatal mental health | Google search for perinatal mental health policies and legislation Review of documents on the National Health Mission, the Ministry of Health and Family Welfare, Federation of Obstetrics and Gynaecology and Literature review | National Health Mission https://nhm.gov.in MoFHW https://main.mohfw.gov.in/?q=Major-Programmes/non-communicable-diseases-injury-trauma/National-Mental-Health-Programme-NMHP New pathways New Hope-National mental health pathway for India https://nhm.gov.in/WriteReadData/l892s/6479141851472451026.pdf Mental Healthcare Act–2017 https://main.mohfw.gov.in/sites/default/files/Mental%20Healthcare%20Act%2C%202017_0.pdf Federation of Obstetrics and Gynaecological Societies of India https://www.fogsi.org/ |
| Perinatal mental health services | Google search for mental health facilities in both states Search of Telangana and Haryana state health websites Informal interviews Literature review | Commissionerate of Health and Family Welfare, National Health Mission, Government of Telangana https://chfw.telangana.gov.in/programmes.html National health mission Haryana https://nhmharyana.gov.in/page?id=214 Interviews with mental health experts, including psychiatrists, psychologists, mental health nurses, researchers and ASHA |
| Perinatal mental health and the community | Informal interviews | Interviews with mental health experts, including psychiatrists, psychologists, mental health nurses, researchers and ASHAs |
| Monitoring and evaluation | Informal interviews | Interviews with mental health experts, including psychiatrists, psychologists, mental health nurses, researchers and ASHAs |
| Informal Interviews | Haryana | Telangana |
| Psychiatrists | Senior psychiatrist in the Faridabad district Hospital, Psychiatrist with 10 years of experience in Rothak | Psychiatrist in Hyderabad with >30 years of experience |
| Psychologists | Psychologist with 15 years of experience | Psychiatrist in Siddipet with >5 years of experience |
| Nurses | 2 psychiatric nurses with >15 years of experience between them | Nurses in Siddipet PHC with >10 years of experience |
| Medical Officers | | Medical officer in Siddipet PHC with >4 years of experience |
| Gynaecologists | | Gynaecologist in Siddipet with >8 years of experience |
| Other | Senior psychiatrist working in Bangalore | District Health Officer in Siddipet with 20 years of experience |

rapport building when dealing with a sensitive topic. These interviews provided an initial understanding of available services and community beliefs. The interviewees are illustrated in Table 1. Interviews were conducted by LM (medical doctor) and research assistants.

### Scoping review

The question guiding the scoping review was: What is the evidence for perinatal mental health policies or policy recommendations in India? The search was limited to journal articles published in English between 1 January 2007 and 3 March 2025. Five databases were searched (Science Direct, Web of Science, PubMed, PsycInfo and Scopus). For a full search strategy, see Supplementary Materials. Inclusion criteria were as follows: (1) Focused on perinatal mental health, (2) Specific policy/health system/legislation recommendations, (3) Focused in India. Exclusion criteria are fully listed in the Supplementary Materials, but papers were excluded if there were no specific policy or system suggestions. For example, an article suggesting that maternal mental health is a priority without positing policy or health system suggestions would be excluded. This was found to be an important distinction in order to identify studies

that actively engage in health system and policy planning. In addition, papers that made clinical practice recommendations for implementation by individual clinicians were excluded. However, if recommendations were made for implementation by a health system as a whole, these papers were included. The literature search yielded 541 articles, which were screened for eligibility, resulting in 82 included publications (Supplementary Materials Figure 2).

## Results

### Relevant context around perinatal mental health

#### Sociodemographic and economic indicators

Relevant data on the socio-demographic and economic situation at a national and state level is presented in Table 2. District-level data is also provided where available. Figures for population size, density and rural population are from the 2011 census (Office of the Registrar General & Census Commissioner, India 2011), as this is the most recent data. Both Haryana and Telangana have better-than-national-average sanitation, with a functioning latrine in 15% more homes in Haryana than the national average (NFHS-5, 2022). Telangana and Haryana have predominantly rural populations, with just under two-thirds of the population living in rural areas, slightly lower than the national average. However, Faridabad district is an exception with a majority urban population. Haryana has a higher population density, with better sanitation, water and electricity provision than seen nationally, along with higher-than-average figures for life-expectancy and female literacy and lower rates of adolescent pregnancy. Telangana has a lower population density than the national Indian average and better sanitation than the national average but below that found in Haryana. Life expectancy and adolescent pregnancy rates in Telangana are similar to the national average, but female literacy is considerably lower. Siddipet and Faridabad have similar levels of sanitation, water and electricity provision, but female literacy rates are higher in the Haryana district. Importantly, Haryana has the lowest proportion of female to male residents (Singh and Hans, 2022), and the sex ratio at birth still remains below the national average. This has led to a shortage of brides in Haryana, and women are being sold into forced marriages or prostitution (Tong, 2022).

Adolescent pregnancy rates have dropped nationally from 7.9% (NFHS-4, 2017) to 6.8% (NFHS-5, 2022) between 2015/16 and 2019/21. Telangana has seen the biggest decrease in adolescent pregnancy from 10.8% (NFHS-4, 2017) to 5.8% (NFHS-5, 2022). Haryana's adolescent pregnancy rate stands at 3.9%. Nationally, nearly a third of women have experienced spousal physical or sexual violence (NFHS-5, 2022). In Telangana, 36.9% of women reported experiencing intimate partner violence when surveyed and 4% during pregnancy. By contrast, the rate of intimate partner violence in Haryana, as measured by the NFHS-5, was 18.2%. Although another study found that 37% of currently married women in Haryana reported ever experiencing domestic violence (Nadda et al., 2018). According to the National Crime Bureau the rate of reported rape per lakh (100,000) population was 12.7 in Haryana (fourth highest rate in the country) and 4.3 in Telangana (National Crime Records Bureau, 2022). The total rate of crimes against women per one lakh of population in 2022 in both Haryana (118.7) and Telangana (117) were among the highest in the country (National Crime Records Bureau, 2022).

#### Maternal health

Key maternal health indicators for India, Telangana and Haryana are illustrated in Figure 1 (Health Management Information System 2020-2021, 2022; NFHS-5 2022; Office of the Registrar General & Census Commissioner, 2022). Telangana has a lower maternal mortality rate than the national average, with Telangana's maternal mortality ratio (MMR) of 43 making it the third best performing state in the country (Office of the Registrar General & Census Commissioner, 2022). By contrast, Haryana's MMR is higher than the national average and over double Telangana's. The Ending Preventable Maternal Mortality (EPMM) initiative (WHO, 2015) has set a number of targets for 2025 for the SDGs to be met. One such target is that 90% of women attend at least four antenatal care visits during their pregnancy (WHO, 2015). As illustrated in Supplementary Table 1, 58.1% (NFHS-5, 2022) of women attended an antenatal clinic at least four times across the country in 2019–2020.

### Policies and legislation around perinatal mental health

At the time of conducting this review, there was no discrete national perinatal mental health policy or plan in place. In addition, a search for a specific perinatal mental health plan within national mental health and maternal health policy produced no results. We conclude that both mental and maternal/child health programmes currently fail to cover perinatal mental health. The search reviewed guidelines and strategies of maternal and mental health bodies. Results are displayed in the Supplementary Materials Table 4. National Health Mission maternal health programmes and the Federation of Obstetrics and Gynaecological Societies of India (FOGSI) guidelines were reviewed, but no guideline on maternal mental health was found. There is clear interest in maternal mental health from maternal health providers, but this has not yet translated into national guidelines or strategy.

The National Mental Health Programme (NMHP) was launched in 1982 (Wig and Murthy 2015) and has since been implemented in most states (National Health Mission 2018; National Health Mission 2019). The District Mental Health Programme (DMHP) was launched in four districts in 1996 as the main implementation arm of the NMHP, to provide community-based mental healthcare. The DMHP has evolved significantly over the years, and is now sanctioned for implementation in 738 districts (Kirpekar et al., 2024). The National Mental Health Policy was released in 2014 (Ministry of Health & Family Welfare, 2014), increasing access to mental health services for vulnerable populations, including a multipart strategy for tackling suicide. The Mental Healthcare Act of 2017 (Government of India 2017) protects the rights of people with mental illnesses and decriminalises suicide. In addition, and of particular relevance to this review, this act mandates joint mother–infant care when a mother is admitted for a mental health crisis.

The scope of this article was to review perinatal mental health policies. However, importantly, other policies and acts will affect maternal mental health either directly or by influencing risk factors contributing to PMHCs. Major key points along the timeline of PMH are mapped below (see Figure 2), and include the launch of the National Mental Health Programme and Act in 1982 and the District Mental Health Programme in 1996, as well as subsequent schemes aiming at providing free, safe delivery in public facilities.

It is important to note that despite legislation, this has not always translated into practice. The dowry remains embedded in religious and cultural traditions; 40–50% of female homicides in India were

**Table 2.** Key socio-demographic and economic indicators on a national, state and district levels for India, Haryana state and Faridabad district, Telangana state and Siddipet district

| SOCIO-DEMOGRAPHIC. AND ECONOMIC INDICATORS | National level – India | | | State level – Haryana | | | District level – Faridabad | | State level – Telangana | | | District level – Siddipet | |
|---|---|---|---|---|---|---|---|---|---|---|---|---|---|
| Literacy | Male | | Female | Male | | Female | Male | Female | Male | | Female | Male | Female |
| | 84.4% | | 71.5% | 91.5% | | 79.7% | | 82.3% | 84.8% | | 66.6% | | 71.0% |
| Sex Ratio of the total population (females per 1,000 males) | 1,020 | | | 926 | | | 890 | | 1,049 | | | 1,062 | |
| Sex Ratio at birth for children born in the last five years (females per 1,000 males) | 929 | | | 893 | | | 955 | | 894 | | | 976 | |
| Homes with sanitation (functioning latrine) | Overall | Rural | Urban | Overall | Rural | Urban | Faridabad | | Overall | Rural | Urban | Siddipet | |
| | 70.2% | 64.9% | 81.5% | 85% | 84.6% | 81.5% | 84.2% | | 76.2% | 72.9% | 81.8% | 84.1% | |
| Homes with clean water supply | 95.9% | 94.6% | 98.7% | 98.6% | 98.2% | 98.7% | 97.9% | | 98.7% | 98.4% | 99.4% | 97.2% | |
| Homes with electricity supply | 96.8% | 95.7% | 99.1% | 99.6% | 99.5% | 99.1% | 100 | | 99.6% | 99.4% | 99.8% | 99.5% | |
| Population size (thousands) | 1,210,855 | | | 25,351 | | | 1810 | | 35,004 | | | 1,013 | |
| Population density (per square km) | 382 | | | 573 | | | 2,442 | | 312 | | | 260 | |
| Population living in rural areas | 69% | | | 65% | | | 20% | | 61% | | | 86% | |
| Life expectancy | Overall | Female | | Overall | Female | | | | Overall | Female | | | |
| | 70.0 | 71.4 | | 69.9 | 73.0 | | | | 70.0 | 71.4 | | | |
| Total Fertility Rate | Overall | Rural | Urban | Overall | Rural | Urban | | | Overall | Rural | Urban | | |
| | 2 | 2.1 | 1.6 | 1.9 | 2.0 | 1.7 | | | 1.8 | 1.7 | 1.8 | | |
| Pregnancy among 15–19 year olds | 6.8% | | | 3.9% | | | 2.8% | | 5.8% | | | 1.0% | |
| Prevalence of intimate partner violence | Overall | During Pregnancy | | Overall | During pregnancy | | | | Overall | During pregnancy | | | |
| | 29.3% | 3.1% | | 18.2% | 1.6% | | | | 36.9% | 4% | | | |

- Population size, density and rural population are from the census 2011
- Life expectancy is from the sample registration system (SRS)-abridged life tables 2016–2020 data
- NFHS for literacy, sex ratios, home with data, fertility rate, total adolescent pregnancy and IPV data
- In the National Family Health Survey-5 (NFHS-5), literacy rates were calculated based on those who have completed at least standard 9 (exam taken during Schooling) or passed a literacy test conducted as part of the survey.
- Prevalence of intimate partner violence is measured in ever-married women aged 18–49.
- Adolescent pregnancy: girls aged 15–19 who were pregnant or already mothers at the time of the survey

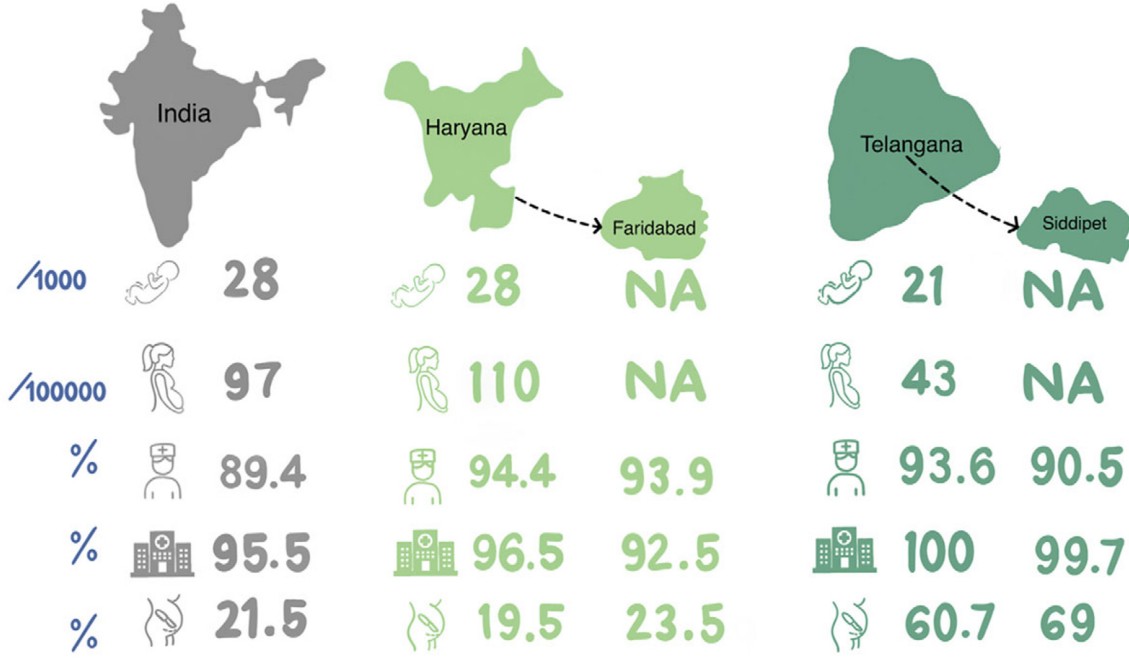

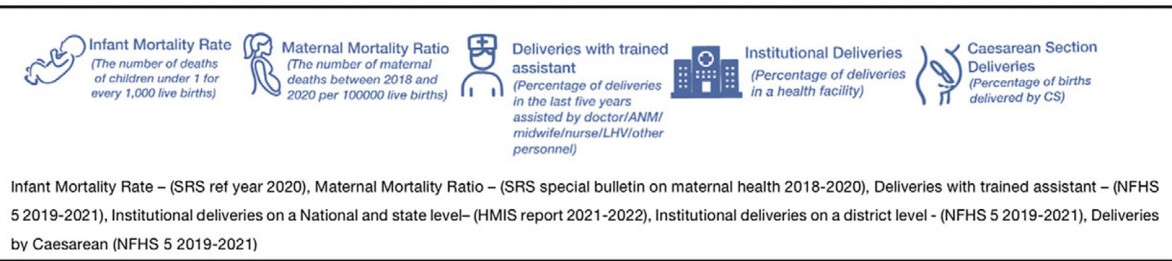

**Figure 1.** Key maternal health indicators for Telangana, Haryana and India.

dowry-related between 1999 and 2016 (United Nations Office of Drugs and Crime, 2019). Sex selective abortions are still a significant problem across the country, as evidenced by the distorted sex ratio at birth, with an estimated 10 million gender biased sex selective abortions reportedly occurring between 1981 and 2005 (Kulkarni, 2007).

### *Perinatal mental health services*

Specialist perinatal mental health inpatient and outpatient services exist at the National Institute for Mental Health and Neurosciences (NIMHANS) in Bangalore (Perinatal Psychiatry NIMHANS), which is home to India's only mother and baby unit. This department provides patient care as well as driving academic work, including perinatal mental health research, as shown in Supplementary Materials Figure 3. It provides assessment and treatment tailored to the individual, including counselling for partners and family members, using multidisciplinary approaches. Perinatal mental health services also exist in other states. Kerala is the only state with a comprehensive perinatal mental health programme, which is implemented through the 'Amma Manasu'(The New Indian Express, 2019) ('mother's mind') programme.

Karnataka has started to integrate mental health into its Reproductive and Child Health programme, with questions on risk factors and early signs of depression and anxiety added to the mother and child protection card (Thayi card) (Ganjekar and Parthasarathy, 2023).

In Haryana and Telangana, general services for mental healthcare exist at the state and district levels. Information from informal interviews with clinicians in Siddipet indicated that perinatal mental health training was delivered in Telangana in 2023. UNICEF and NIMHANS are running a project aiming to enhance maternal nutrition and improve the mental health of pregnant women (UNICEF, 2023). Working with the Government of Telangana, weight tracking, nutritional counselling, anaemia prevention and mental health screening will be integrated into routine antenatal check-ups (UNICEF, 2023). There is no mention of any perinatal mental health plan on the Haryana Health Department website (Health Department Haryana), and services are limited currently.

An overview of the availability of mental healthcare professionals, inpatient and outpatient facilities on a district level is displayed in Table 3.

Telangana's largest centre for mental healthcare is the Institute of Mental Health in Hyderabad. A general mental healthcare centre

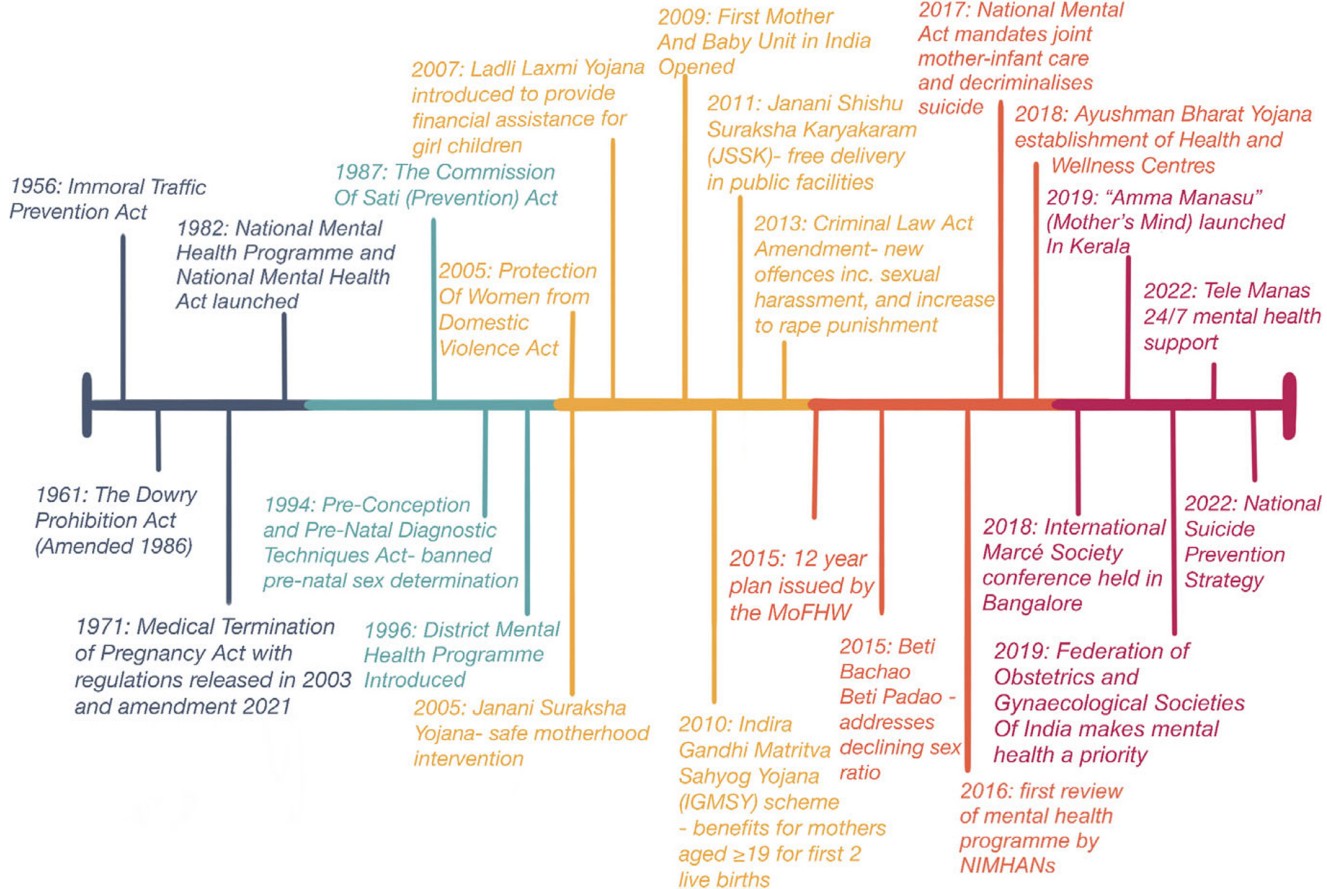

**Figure 2.** Relevant policies and major milestones affecting perinatal mental health in India.

exists in Siddipet, at the Government General Hospital, which has a 30-bed psychiatric ward and day clinic, and treats referrals from primary health care and walk-ins. Importantly, primary healthcare doctors do not treat mental health cases, but refer on to tertiary centres for treatment. Perinatal mental health orientation training was provided to Primary Health Care (PHC) doctors and Maternal and Child Health (MCH) supervisors (by NIMHANS in October 2023) for all districts in Telangana, including Siddipet. Auxiliary Nurse Midwives (ANMs) will be trained to complete mental health screening questionnaires with all pregnant women. Then, PHC doctors will review the score and refer to tertiary hospitals for treatment. Comments from informal interviews informed this section.

The State Mental Health Institute, Pandit Bhagwat Dayal Sharma University of Health Sciences in Rohtak, is the major mental health institute for the state of Haryana. In addition, the Department of Psychiatry Postgraduate Institute of Medical Education and Research, Chandigarh was one of the earliest general psychiatric units and runs satellite clinics in Haryana. There are no specific PMH services in Haryana, but cases can be seen in the District Hospital in Faridabad. Travel to district hospitals in both states can be challenging and expensive for women and their families. For instance, it can take around 2 hours from the furthest village in Faridabad to travel to the District Hospital by car, and there are no feasible public transport options. Private clinics and services exist. As part of the general mental health programme, awareness raising sessions are run for the ANMs, staff nurses, medical officers of the PHCs and Community Health Centres (CHCs). During these sessions information is given about general mental health, signs and symptoms of mental illness, prevention strategies and stigma. Accredited Social Health Activists (ASHAs) are also given a day of annual training on antenatal and postnatal mental health, to allow them to recognise symptoms, provide basic counselling and identify those who need further help. Basic psychiatric medicine (mostly antidepressants are available at the PHCs and CHCs). District counselling services exist in Haryana ('14th Common Review Mission' 2021). Again, data from informal interviews informed this section.

*Perinatal mental health policies, systems and legislation: Literature review*

The review of perinatal mental health policies, systems and legislation included 82 publications. An overview of these publications, their design, focus, location and policy/systems commentary is provided in Table 4. There were 26 prevalence studies, and 33 studies looked at risk factors or determinants of perinatal mental health conditions. Six studies focused on the Mental Health Care Act, and 12 studies were primarily focused on health systems or policies. Twenty-Two of the studies were performed at the national level, and 60 were performed at the state level, primarily in South India (Supplementary Materials Figure 3).

The evidence for specific policy/system recommendations varied: for some recommendations there was empirical evidence supporting feasibility and efficacy; other comments arose from the practical experience of a broad range of professionals in the field. Figure 3 illustrates some of the key recommendations. There is a broad consensus that mental healthcare for women during

**Table 3.** Available district-level healthcare in Faridabad district (Haryana) and Siddipet district (Telangana)

| Obstetrics & gynaecology services | Faridabad district (Haryana) | Siddipet district (Telangana) |
|---|---|---|
| Obstetrician & Gynaecologist | 8 | 6 (as well as 4 postgraduate students) |
| Paediatricians | 2 | 5 (as well as 4 postgraduate students) |
| Staff nurse | 14 | 250 |
| Anaesthesiologists | 2 | 4 |
| Compounder | 0 | 15 |
| Housekeeping | 10 | 150 |
| Total beds | 55 | 500 |
| OP visits/month | 50 | Up to 20,000 |
| Monthly delivery rate | 150–160 | 500 |
| %Caesarean section deliveries | Roughly 5% | 50% |
| **Available Mental Health Human Resources** | | |
| Available Human Resources | Faridabad district (Haryana) | Siddipet district (Telangana) |
| General perinatal mental health workers | No | No |
| General doctors | Yes | Yes |
| Health Officers | Yes | Yes |
| Midwives | Yes | Yes |
| Degree nurses | Yes | Yes |
| Diploma nurses | Yes | Yes |
| Neurologist | No | Yes |
| Psychiatrist | Yes | Yes |
| Psychiatric practitioners | Yes | Yes |
| Psychiatric nurse | Yes | No |
| Psychologist | Yes | Yes |
| **Perinatal mental health inpatient care** | | |
| Perinatal inpatient care | Faridabad | Siddipet |
| Specialist perinatal mental healthcare | No | No |
| Nearest general inpatient facility | District Hospital | General mental healthcare in Government General Hospital in Sidipet (GGH). This has a ward with 30 beds. Also Institute of Mental Health Hyderabad |
| Is alcohol detoxification offered? | Yes – District Hospital has a dedicated de-addiction programme, and inpatient facilities at the district hospital are available for de-addiction | Yes – there is a separate drug centre at the Institute of Mental Health, Hyderabad, and facilities in Siddipet |
| **Perinatal mental health outpatient care** | | |
| Specialist Perinatal mental health outpatient care | Faridabad | Siddipet |
| What are the nearest facilities? | District Hospital – can take about 2 h from the furthest village to reach this hospital | Government General Hospital in Siddipet and the District Hospital in Gajwel |
| Referral systems in place? | Community psychiatric nurse conducts home visits to pregnant women. They can identify those who need further support. General doctors are asked to identify potential cases and refer them to the District Hospital. ASHA workers have also been trained to refer women for further treatment and diagnosis. | Screening questionnaires on ANC cards – scores are reviewed by PHC doctors who refer onwards to GGH/tertiary hospitals for treatment. In addition, private healthcare providers and obstetricians can make referrals. Women can also self-refer to outpatient services. |

*Note:* Informal Interviews were conducted to complete this table.

pregnancy and in the postnatal period should be integrated into routine maternal health services (Mariam and Srinivasan, 2009; Prost et al., 2012; Bagadia and Chandra, 2017; Shidhaye et al., 2017; Rathod et al., 2018; Sheeba et al., 2019; Amipara et al., 2020; Babu et al., 2020; Ransing et al., 2020; Ransing et al., 2020a; Behl, 2021, 2023; Ransing et al., 2021; Singla et al., 2021; Kalra et al., 2021, 2022; Kukreti et al., 2022; Nisarga et al., 2022; Priyadarshini et al., 2023; Thomas et al., 2023; Behl et al., 2024, 2025; Handa et al., 2024;

**Table 4.** List of included studies in this review

| Year | Title | Location | Focus of study | Type of study | Comments on Policy/systems |
|------|-------|----------|----------------|---------------|----------------------------|
| **Intervention studies** | | | | | |
| (Ransing et al. 2021) | Development of a brief psychological intervention for perinatal depression (PND) | National | Development of a scalable model of care for PND | Literature review and expert opinion discussion | Urgent need to address the burden of PND and to reduce the treatment gap in India. Screening accompanied with interventions is associated with a significant increase in use of perinatal mental healthcare services.<br>WHO recommends a stepped care model with self-help interventions and non-specialist interventions to reduce the gap.<br>Need for an integrated model of care aimed at training primary health care workers and a multipronged strategy.<br>Consensus reached to develop an auxiliary nurse midwife-based stepped care model consisting of care, training and referral services for PND. |
| (Fuhr et al., 2019) | Delivering the Thinking Healthy Programme for perinatal depression through peers: an individually randomised controlled trial | Goa | The effectiveness and cost-effectiveness of the Thinking Healthy Programme (THP) | Single blind, Individually randomised Controlled trial | This programme had moderate effects On remission from perinatal depression, was relatively cheap to deliver and was cost-saving. Thus, it could be used as part of a stepped care approach.<br>This may also give rise to a new cadre of healthcare workers – Sakhis who are able to tackle the treatment gap of maternal depression. |
| (Singla et al., 2021) | Multiple mediation analysis of the peer-delivered Thinking Healthy Programme for perinatal depression: findings from two parallel randomised controlled trials | Goa | Examine three mediators of the Thinking Healthy Programme | Two parallel RCTs | Interventions should target both social support and patient activation levels.<br>To achieve integration of mental health into other services, a stronger emphasis on mother–child attachment may be required. Generalisability of the THPP. Peer-delivered interventions have the potential to be more feasible than other interventions and may result in better adherence, especially among those who are more socio-economically disadvantaged and isolated from the healthcare system. |
| (Supraja et al., 2016) | Suicidality in early pregnancy among antepartum mothers in urban India | Bangalore | Prevalence and predictors of suicidality | Cross-sectional study nested within a prospective cohort study | Training obstetric healthcare staff to identify women at risk and provide support and referral should be an important initiative. |
| (Shiva et al., 2021) | A virtual course in perinatal mental health for healthcare professionals | Bangalore | Explore the role of a virtual certificate course in PMH | Educational intervention or pre/post study | Encouraging feedback indicates that virtual courses could help enhance competency in healthcare professionals to identify and treat PMH.<br>Strong need to develop expertise in PMI in LMICs, and this virtual course could be provided globally. Training health professionals is necessary, and this format lends itself to scale-up.<br>Involve learners from multidisciplinary backgrounds, keep courses interactive and include case-based and peer learning. |

**Table 4.** (*Continued*)

| Year | Title | Location | Focus of study | Type of study | Comments on Policy/systems |
|------|-------|----------|----------------|---------------|----------------------------|
| (Vaiphei et al., 2023) | Formation of a stakeholder group of women with a lived experience of Postpartum Psychoses (PP) – Experience from a perinatal psychiatry service in India | Bangalore | Process of forming a stakeholder group for PP | Qualitative | Value of participatory approaches in healthcare systems. Patients and their caregivers can be involved in health system policy, planning, service monitoring and research – which can strengthen the mental health system. Can lead to more acceptable and accessible health services, and improve the responsiveness of services. Indeed, involvement of stakeholders is increasingly being recognised as critical to service development. Women with PP and their caregivers can be involved as stakeholders in mental health decision-making – it appears feasible in an LMIC setting and should be encouraged. Need to ensure adequate gender representation and urban/rural representation. Suggestions stemming from people's lived experiences are important in understanding their unique service needs as well as helping to refine research questions. |
| (Thomas et al., 2023) | Feasibility of training primary healthcare workers to identify antenatal depression | Bangalore | Feasibility of training PHC workers in an urban antenatal clinic in screening pregnant women with a questionnaire | Pre-post design | Task shifting by training primary healthcare workers in screening could help in identifying and referring more women with antenatal depression at an early stage. Training healthcare workers (nurses, nurse assistants and ASHAs) in antenatal clinics is feasible and can increase the identification rate of depressive symptoms in pregnant women. The training improved staff's knowledge, perceived skills and self-efficacy – which may make them more prepared for task shifting. There was an increase in the number of women identified with depressive symptoms once a structured screening was initiated. The MHGap intervention guide directly targets knowledge. Some women with moderate depressive symptoms declined referral for fear of being perceived negatively for seeking help for their mental illness. Need for education in the community addressing mental health disorders and stigma as well as providing interventions for women with symptoms and integrating mental health services into routine ANC. Policy decisions should be taken to incorporate mandatory screening in antenatal care settings along with interventions based on illness severity. |
| (Prabhu et al., 2025) | Effectiveness of psychosocial education programme on postnatal depression, stress and perceived maternal parenting self-efficacy | Manipal | Effect of a psychosocial education programme on postnatal depression, maternal stress and self-efficacy | Randomised Controlled Trial | The programme was effective in reducing postnatal depression and stress related to pregnancy and childbirth. Preventive interventions might be more effective if conducted during |

**Table 4.** (*Continued*)

| Year | Title | Location | Focus of study | Type of study | Comments on Policy/systems |
|---|---|---|---|---|---|
| | among pregnant women in South India | | | | pregnancy due to the high incidence of PPD in the first few months postpartum.<br>Nurses in a maternity unit could be trained to conduct a brief psychosocial education programme, which can be incorporated into routine care.<br>This intervention was simple, low-cost cost and effective and can support continuity of care. |
| (Shidhaye et al., 2016) | Development and piloting of a plan for integrating mental health in primary care in Sehore district, Madhya Pradesh, India | Madhya Pradesh | Operationalise the WHO Mental Health Gap Action Plan and design an integrated mental healthcare plan (MHCP) | Situational analysis, qualitative study | The role of psychiatrists needs to undergo a fundamental paradigm shift, and they should play a major role in tasks such as designing mental healthcare programmes that can be delivered by community mental health teams, as well as providing supportive supervision to these teams.<br>Focus shifts from clinical interventions to systems strengthening. |
| (Lakshminarayanan et al., 2020) | Delivery of perinatal mental health services by training lay counsellors using digital platforms | Karnataka, Maharashtra, Bihar, Jharkhand | Remote training of people with no prior knowledge surrounding perinatal mental health to recognise mental illness and deliver psychosocial services | Feasibility study | Digital training is a viable alternative to face-to- face training in situations with a limited budget and other constraints.<br>There is a compelling requirement to address the maternal mental health gap and this initiative has generated evidence in favour of cost-effective training for psychosocial treatment in less-resourced settings. |
| (George et al., 2020) | Effectiveness of a group intervention led by lay health workers in reducing the incidence of postpartum depression in South India | Kerala Tamil Nadu | Low-intensity group intervention led by lay workers during the antenatal period | Parallel group design with both positive and negative controls | This intervention, with its possibility of reducing postnatal depression, could be scaled up for testing/ intervention.<br>Need to develop locally relevant, innovative and feasible strategies for the prevention of postnatal depression.<br>Innovative techniques, such as smartphone technology use could be employed.<br>High levels of satisfaction among the recipients confirm the feasibility of effective task shifting in similar settings. |
| (George et al., 2016) | Antenatal depression in coastal South India- Prevalence and risk factors in the community | Kerala and Tamil Nadu | Prevalence of antenatal depression in women from a rural coastal background and associated risk factors | Cross-sectional community-based study | Need to develop strategies for recognition and appropriate intervention in the context of locally relevant risk factors.<br>Need for a universal screening programme and treatment strategies for antenatal depression. Policy makers must prioritise addressing emotional disorders within programmes which seem to focus preferentially on reducing maternal and infant mortality.<br>Shorter, time-efficient and accurate screening methods that can be employed by healthcare workers at the primary care level during obstetric/infant-related appointments must be developed.<br>Management of depression may |

(*Continued*)

**Table 4.** (*Continued*)

| Year | Title | Location | Focus of study | Type of study | Comments on Policy/systems |
|---|---|---|---|---|---|
| | | | | | require a stepped care approach. Multisectoral response is needed, addressing poverty reduction, social protection, violence prevention, education and gender disadvantage. |
| (Fellmeth et al., 2021) | Perinatal mental health in India: protocol for a validation and cohort study | Karnataka and Himachal Pradesh | Outline a study on PMH which will aim to improve understanding of CMD in women of reproductive age | Protocol | Screening for PMI should occur in maternal health and obstetric settings by healthcare workers, including nurses, midwives and primary care doctors. Screening tools must be simple, reliable and quick to administer as well as locally validated and culturally appropriate. They must have established cut-offs to facilitate stepped care. |
| (Kukreti et al., 2022) | Stepped Care Model for Developing Pathways of Screening, Referral and Brief Intervention for Depression in Pregnancy: a Mixed-Method Study from Development Phase | Delhi and Ratnagiri | Develop and implement a stepped care model | Mixed Methods | Stepped care models using existing antenatal healthcare staff or peers for screening maternal depression and providing brief intervention have shown promising results across South Asian countries. Lay health workers can deliver such models. The Brief psychological intervention for perinatal depression (BIND-P) stepped care pathway is a systematic path for the integration of mental health principles in general healthcare settings. Preliminary result shows the model is feasible, acceptable, and there is potential for upscaling for integration into reproductive child health programmes. |
| (Tripathy et al., 2010) | Effect of a participatory intervention with women's groups on birth outcomes and maternal depression in Jharkhand and Orissa, India: a cluster-randomised controlled trial | Jharkhand Orissa | Effects of a low-cost participatory intervention on neonatal mortality and maternal mental health | Randomised Controlled trial | This programme led to reduction in depression among women in the third year of its running and a 32% reduction in neonatal mortality rate. Such an intervention could be supported by the government and could strengthen the National Rural Health Mission's mandate for communities being at the centre of its programme. |
| **Prevalence and risk factor studies** | | | | | |
| (Kalra et al., 2021) | Prevalence and determinants of antenatal common mental disorders among women in India: a systematic review and meta-analysis | National | Prevalence of antenatal CMDs and their determinants among women in India (risk factors and protective factors) | Systematic review and meta-analysis | There is an urgent need for locally developed policies and programmes, as well as integration of preventative and early intervention mental health programmes into maternal care. Lack of availability of synthesised local evidence of perinatal mental health morbidity in India; most studies found were on populations in South India and based on urban populations. |
| (Kalra et al., 2022) | Burden of severe maternal peripartum mental disorders in low- and middle-income countries: a systematic review | National | Prevalence of severe peripartum mental disorders | Systematic review | Psychosocial disadvantage and gender related risk factors have a negative influence on health-seeking behaviours and reduce access to treatment. There are far more negative consequences of severe mental Illness (SMI) in LMIC settings. |

**Table 4.** (*Continued*)

| Year | Title | Location | Focus of study | Type of study | Comments on Policy/systems |
|---|---|---|---|---|---|
| | | | | | These factors make postpartum psychosis a public health priority. The authors strongly support the integration of mental healthcare into routine maternal care in these settings. |
| (Kedare et al., 2024) | Mental health and well-being of women (menarche, perinatal and menopause) | National | Mental health and well-being considerations and determinants of women in each phase | Chapter | Workplace support should be available – all institutions should have a good environment for pregnant and lactating women. Mental health services and maternal and child services should overlap with more contact and interaction between both at each obstetric visit. Greater education about symptoms of a mental health problem – likely best done by an MCH service. The government or DMHP needs to train healthcare providers like ASHAs, community midwives, community social workers and those working in MCH in PMI. Healthcare workers' attitudes and behaviours are important in promoting mental health. Mindfulness training can be incorporated into community health programmes. |
| (Rajeev et al., 2024) | India's Silent Struggle: A Scoping Review on Postpartum Depression in the Land of a Billion Mothers | National | Evidence on the awareness and prevalence of PPD in India | Scoping review | Most studies emphasised the need to establish support systems and strengthen the existing healthcare system. Hospitals need to develop referral systems and establish postpartum clinics. Urgent need for more geographically inclusive studies in India to represent samples across states. Highlights the importance of accessible services related to PPD. Education of healthcare service providers to identify PPD early. An online or offline screening system should be made available so that routine screening can be administered. Establish a knowledge repository in English and local languages for women. Establish specialised postpartum clinics attached to paediatric departments. |
| (Mariam and Srinivasan, 2009) | Antenatal psychological distress and postnatal depression: A prospective study from an urban clinic | Bangalore | Prevalence and risk factors in the development of postpartum depression | Prospective cohort study | Antenatal psychological distress was a risk factor in the development of PPD, hence a need for routine screening of psychological distress during antenatal visits. Mental healthcare must be made part of routine antenatal services to facilitate early identification and treatment of psychological distress during pregnancy. |
| (Kishore et al., 2018) | Life events and depressive symptoms among pregnant women in India: Moderating role of resilience and social support | Bangalore | Understand the association between life events and perinatal depression, and whether that is moderated by resilience and support | Prospective cohort study | Life events predicted depression during pregnancy and the relationship was moderated by social support but not resilience. Since it is not always possible to prevent adverse life events, appropriate intervention programmes need to be planned to enhance adequate social support to deal with their impact. |

(*Continued*)

**Table 4.** (*Continued*)

| Year | Title | Location | Focus of study | Type of study | Comments on Policy/systems |
|---|---|---|---|---|---|
| | | | | | Inclusion of assessment of life events in the preceding year as part of comprehensive psychosocial screening, and consider interventions for enhancing social support for those who report them. |
| (Sheeba et al., 2019) | Prenatal depression and its associated risk factors among pregnant women in Bangalore: A hospital-based prevalence study | Bangalore | Prevalence of prenatal depression and its associated risk factors among pregnant women in Bangalore | Cohort study | Obstetric practice should include screening and diagnosis of prenatal depression as part of routine antenatal care in low and middle-income countries. |
| (Mahale et al., 2021) | A study of Postpartum depression and its risk factors in a tertiary Hospital in India | Karnataka | Prevalence of PPD women attending a tertiary hospital and associated risk factors | Hospital-based Cross-sectional | Risk factors can be identified during routine antenatal visits (low level of education, low socio-economic status, age at marriage, parity, obstetric complication, previous marriage, male baby preference, neonatal complications, history of psychiatric disorder, living in joint families, lack of family support, history of marital conflict and alcohol abuse in the husband) and hence these issues must be addressed by healthcare providers so that at risk women can be identified for early follow up. Incorporate mental healthcare programmes into the Maternal Health Care system. |
| (Nisarga et al., 2022) | Social and obstetric risk factors of antenatal depression | Karnataka | Modifiable social and obstetric risk factors of antenatal depression | Cross-sectional study | Focus must be on screening for depression during pregnancy and providing adequate psychosocial support with treatment for depression. This study found antenatal depression to be very common, thus screening should be included as a routine part of antenatal care and an attempt to screen must be at least made in high-risk individuals (urban residents, and those affected by fear of labour, intimate partner violence, poor relationship with in laws and parents). Antenatal depression is multifactorial in origin and requires a multifactorial approach to prevention and treatment with efforts at the community level. Integrating mental health within maternal and child services should be adopted at a community level. |
| (Basu et al., 2021) | Postpartum depression burden and associated factors in mothers of infants at an urban primary health centre in Delhi, India | Delhi | Burden of PPD and associated factors in women with an infant | Cross-sectional | Regular mandatory screening for PPD is needed in primary healthcare facilities for an extended period post-childbirth. |
| (Kukreja et al., 2012) | Persistent postnatal depression after preterm delivery | New Delhi | Prevalence of poor psychological well-being and depression at 6 weeks post-delivery | Prospective cohort study | Higher incidence of postnatal psychological distress and depression at 6 weeks after delivery among mothers with preterm infants. All mothers who had a preterm delivery should be screened for |

(*Continued*)

**Table 4.** (*Continued*)

| Year | Title | Location | Focus of study | Type of study | Comments on Policy/systems |
|---|---|---|---|---|---|
| | | | | | features of psychological distress during the early postnatal period to ensure early detection and management of the condition. |
| (Bachani et al., 2022) | Anxiety and depression among women with COVID–19 infection during childbirth – experience from a tertiary care academic centre | New Delhi | Prevalence of depression and anxiety in women testing positive for COVID–19 and associated factors | Cross-sectional study | Need for increased screening for common mental illnesses with identification of associated risk factors.<br>Strong need for liaison between obstetricians and mental health professionals.<br>If worries about the effects of COVID–19 are out of proportion, referral services should be made available.<br>Addressing mental health concerns early on prevents long-term consequences. |
| (Mahapatro et al. 2022) | Domestic violence during pregnancy as a risk factor for stress and depression: The experience of women attending ANC at a tertiary care hospital in India | New Delhi | Association of domestic violence (DV) with stress and depression during the first trimester | Mixed Methods | The strong association between DV victimisation around the time of pregnancy and the likelihood of a mother exhibiting stress and depressive symptoms reinforces the need to conduct routine screening during pregnancy to identify women with a history of domestic violence/currently experiencing domestic violence.<br>Call for significant attention in the planning of health education programmes for all staff to identify and screen the risk of DV and associated stress and depression, as well as for the implementation of holistic interventions.<br>Initial rapport building with trust and bonding, customised care and a patient-centric approach.<br>Necessity for discrete screening and creation of safe spaces within the institutional healthcare system, and responsive and reactive caregiving |
| (Arora and Aeri 2021) | Association between high pre-pregnancy body mass index and antenatal depression: A study among pregnant women of upper socio-economic strata in North-West Delhi, India | Delhi | Prevalence of AD among normal weight, overweight and obese pregnant women | Observational study part of a longitudinal study | This study has provided evidence for the hidden burden of AD and its association with pre-pregnancy BMI even among women of higher socio-economic status.<br>A holistic approach needs to be adopted by healthcare professionals.<br>Regular screening of pregnant women for depression must be conducted, as well as lifestyle advice.<br>This study can provide an impetus for the development of interventions aimed at addressing maternal mental health |
| (Kumari and Basu, 2024) | Postpartum depression and its determinants: A Cross-sectional study | Delhi | Prevalence and determinants of PPD among women reporting to secondary care | Cross-sectional study | Prioritising birth-preparedness and strengthening health system screening protocols may prevent and reduce the effects of PPD.<br>Increased focus on training and sensitisation of health professionals for screening with questionnaires.<br>National health programmes in India should include strategies and interventions for early recognition and treatment of PPD.<br>Integrate evidence-based |

**Table 4.** (*Continued*)

| Year | Title | Location | Focus of study | Type of study | Comments on Policy/systems |
|---|---|---|---|---|---|
| | | | | | behavioural modifications within essential healthcare packages for mothers and infants. Expand the role of telemedicine. Promote the availability of antidepressants through trained primary care providers. |
| (Shidhaye et al., 2017) | Association of gender disadvantage factors and gender preference with antenatal depression in women: a cross-sectional study from rural Maharashtra | Maharashtra | Prevalence of probable antenatal depression and association of marital and gender disadvantage factors and child gender preference | Cross-sectional | The maternal mental health agenda should be linked to addressing broader social development goals and gender empowerment. Decision-makers involved in maternal and child health and mental health must collaborate with each other to integrate mental health into primary care. Nurses and other para-medical staff in PHCs can be trained to screen for antenatal depression. |
| (Chainani, 2021) | Incidence of postpartum depression in a tertiary care hospital in Navi Mumbai amid the COVID–19 Pandemic | Maharashtra | Incidence of postpartum depression (PPD) during the COVID–19 pandemic and identify sociocultural triggers | Cross-sectional | COVID–19 and worldwide lockdowns are causing widespread psychological problems, and PPD is quite common in this part of the country. Early identification of PMI can be lifesaving. A support system, plenty of transparent medical advice, and compassion must be applied to doctor visits. Doctors also need to familiarise themselves with PPD scoring systems and also clinical symptoms to be able to reassure and diagnose women ahead of severe maternal morbidity. |
| (Doke et al., 2021) | Assessment of difference in postpartum depression among caesarean and vaginally delivered women at 6 week follow up in hospitals in Pune District, India: an observational cohort study | Maharashtra | Compare the proportion of PPD among women who had CS or vaginal delivery and associated socio-demographic risk factors | Observational cohort study | Odds of PPD are higher in women who had a CS, and women aged less than 25 had higher odds. Young women, particularly those who had a CS delivery, should be screened 6 weeks after delivery. Universal assessment and screening of women (particularly young women and those who had a CS delivery) should be carried out at 6 weeks postpartum in all hospitals using EDPS through nurses or social workers. |
| (Singh et al., 2023) | Association between gestational diabetes mellitus and postpartum depression among women in Eastern India: a cohort study | Odisha | Association between GDM and PPD | Prospective cohort study | Women with gestational diabetes mellitus (GDM) were at higher risk of developing PPD, suggesting an at-risk approach should be implemented for screening. After appropriate training, front-line healthcare workers can screen during follow-up visits for GDM and as part of the home-based newborn programme. Women can be educated about the need to approach health facilities if they develop any signs or symptoms of PPD. This multipronged provider and patient-based approach, along with |

(*Continued*)

**Table 4.** (*Continued*)

| Year | Title | Location | Focus of study | Type of study | Comments on Policy/systems |
|---|---|---|---|---|---|
| | | | | | linkage of front-line healthcare workers, can identify those at risk of PPD and offer interventions when needed. |
| (Kar et al., 2024) | Family and facility care variables attributing to postnatal depression among women in two tribal dominated districts of Odisha: Log model analysis to suggest intervention | Odisha | Postnatal depression and its attributes in the early postpartum period | Cross-sectional study | Robust healthcare alone will not be able to address the holistic health needs of childbearing women. Ingrained familial perceptions and structures are the barriers to success. Family stressors compound poor mental health, and counselling of the family as a whole is needed to achieve sound mental health for women postnatally. The EPDS is a simple, cost-effective tool that can be integrated into health assessment systems. |
| (Prost et al., 2012) | Predictors of maternal psychological distress in rural India: a cross-sectional community-based study | Jharkhand | The prevalence and predictors of psychological distress as a proxy for common mental disorders among mothers | Cross-sectional community-based | Scaling up interventions with lay/community health workers or group interventions. Screening with simple tools could be integrated with maternal healthcare, especially for those who are high risk. |
| (Srinivasan et al., 2015) | Assessment of the Burden of Depression During Pregnancy Among Pregnant Women Residing in the Rural Setting of Chennai | Chennai | Burden of antenatal depression and associated risk factors | Cross-sectional study | True estimates of the prevalence of antenatal depression, as well as risk factors, can provide evidence for research and policy development. Integrating screening into routine antenatal care can reduce untreated antenatal depression and can be carried out by trained non-mental health professionals. |
| (Goyal et al., 2020) | Psychiatric Morbidity, Cultural Factors and Health-Seeking Behaviour in Perinatal Women: A Cross-sectional Study from a Tertiary Care Centre of North India | North India | Prevalence of psychiatric morbidity as well as cultural factors influencing illness, help-seeking behaviour and barriers to care | Cross-sectional study | Psychosocial stressors, marital conflict and past history of psychiatric illness were significantly higher in women with perinatal mental illness. So, if psychological support could be provided to these women on time, it could reduce the occurrence of depression. Religion and social support were major coping strategies, and stigma and financial problems were major barriers to help seeking with less than a third reporting medical help. Cultural aspects need to be addressed in planning, prevention and treatment programmes. |
| (Jha et al., 2021) | Prevalence of Common Mental Disorders among pregnant women – evidence from a population-based study in rural Haryana, India | Ballabgarh – Haryana | Burden of common mental disorders (CMDs) among pregnant women in rural Haryana | Community-based Cross-sectional | Need for the integration of screening of CMDs into routine antenatal care. Recommend the integration of mental healthcare with routine antenatal services at a primary care level. Primary care physicians and ANMs can be trained to use screening tools like PRIME MD PHQ–9 to aid in detecting CMDs at an early stage and timely referral to a specialist. This should also be supported by awareness-generating activities at a community level to reduce stigma. |
| (Sarkar et al., 2022) | The unheard parental cry of a stillbirth: fathers and mothers | Chandigarh | Prevalence of depression among parents after a stillbirth | Prospective cohort study | Advocate for the development of health policies for mental health screening in couples following a |

(*Continued*)

**Table 4.** (*Continued*)

| Year | Title | Location | Focus of study | Type of study | Comments on Policy/systems |
|---|---|---|---|---|---|
| | | | | | stillbirth.<br>Need to bring changes in hospital policies to promote psychotherapy to bereaved parents.<br>Possible value of web-based mental health services, including informative websites, self-help groups, virtual counselling services and automated therapy programmes but need for systematic evaluation.<br>Need of the hour: set up protocols for tailor-made bereavement services to offer couples suffering stillbirths bereavement care and counselling, and health professionals should screen these parents for MH problems. |
| (Jelly et al., 2021) | Impact of the COVID–19 Pandemic on the psychological status of pregnant women | Uttarakhand | Psychological impact of COVID–19 on pregnant women | Cross-sectional survey | Early identification of high-risk women is important in guiding and planning to reduce the complications associated with maternal psychological stress.<br>It is essential to assess the psychological impact of stressful events such as the pandemic on pregnant women and plan effective strategies to reduce their impact. |
| (Rathod et al., 2018) | Characteristics of perinatal depression in rural central India: a cross-sectional study | Madhya Pradesh | Epidemiological features of depression among perinatal women | Secondary analysis of cross-sectional surveys | The high-risk groups highlighted in this study (18–22 yrs., higher disability, facility attendees) should be made a priority for mental healthcare.<br>Mental health services should be integrated into maternal healthcare platforms. Use the maternal health infrastructure as a foundation for the provision of maternal mental healthcare.<br>Use economic modelling to predict the cost–benefit with different service models.<br>Service planners must recognise that treating depression is likely to have a partial effect on reducing suicidal ideation.<br>With adequate training and supervision, nurses and ASHA workers can be capacitated to screen vulnerable mothers and provide psychosocial interventions as indicated.<br>Evidence generated by this study and others must be used by policy-makers to prioritise mental health services for mothers alongside maternal and child services. |
| (Amipara et al., 2020) | A study on postpartum depression and its association with infant feeding practices and infant nutritional status among mothers attending the Anganwadi centres of Valsad District, Gujarat, India | Gujarat | Sociodemographic risk factors associated with PPD and its possible association with infant feeding practices and infant nutritional status | Cross-sectional | Integration of maternal mental health into existing maternal and child health services<br>Screening procedures for PPD and counselling should be introduced at a field level.<br>Early and timely preventative measures, referral and treatment are important in decreasing maternal suffering and preventing the harmful effects on child growth. |

**Table 4.** (*Continued*)

| Year | Title | Location | Focus of study | Type of study | Comments on Policy/systems |
|---|---|---|---|---|---|
| (Prabhu et al., 2022) | Prevalence and associated Risk Factors of Antenatal depression among Pregnant women attending Tertiary Care Hospitals in South India | Manipal | Prevalence and risk factors associated with AD in South India | Cross-sectional | Screening for prenatal depression as part of routine pregnancy care and prompt treatment. Regular antenatal care should include screening and diagnosis of antenatal depression (AD). Healthcare professionals must create educational programmes to assist expectant mothers and offer psychological prenatal care assistance. |
| (Priya et al., 2024) | Prevalence and risk factors of postpartum depression in Sub-Himalayan region | Himachal Pradesh | Prevalence of PPD and its associated risk factors in a hilly region in Himachal Pradesh | Hospital-based cross-sectional study | High prevalence of PPD was found in this region, and, therefore, routine screening for PPD should occur at 6 weeks postpartum. Planning effective public health measures towards PPD requires a precise estimation of the burden of this disease and its risk factors. |
| (Easwaran et al., 2025) | Depression, anxiety and stress among HIV positive pregnant women during the COVID–19 pandemic: a hospital-based cross-sectional study in India | Andhra Pradesh | Prevalence and factors associated with DAS (perinatal depression, anxiety and stress) among HIV positive women | Cross-sectional study | Integrate maternal mental healthcare into antenatal care in all primary healthcare settings. Telehealth services can meet the needs of antenatal and HIV care to build confidence among rural HIV-positive pregnant women to sustain their mental health and wellbeing. Design educational programmes targeting COVID–19 stressors and implement them at the institutional level to improve the mental health of HIV-positive pregnant women. Holistic interventions delivered by a multidisciplinary team, including a clinical psychologist, obstetrician and HIV prevention and treatment specialist. Pandemic preparedness involves not only the development of healthcare infrastructure but also handling mental health challenges in the public and special populations. |
| (Scott et al., 2020) | Multidimensional. Predictors of common mental disorders among Indian mothers of 6- to 24-month old children living in disadvantaged rural villages with women's self-help groups: A cross-sectional analysis | Jharkhand, Madhya Pradesh, West Bengal, Odisha, Chhattisgarh | Risk factors for CMD across multiple aspects of life | Cross-sectional survey data | Practitioners and policymakers should aim to improve food security, economic wellbeing and social capital, such as that created through self-help group (SHG)membership, to improve maternal mental health. |
| (Badiya et al., 2020) | Identification of clinical and psychosocial characteristics associated with perinatal depression in the south Indian population | Andhra Pradesh Karnataka | Prevalence and characteristics associated with perinatal depression | Longitudinal observational study | To mitigate the risks associated with these characteristics (urban living, recent adverse life events, irregular menstrual history and pregnancy complications) it is important to implement appropriate intervention programmes for pregnant women and new mothers. As India is culturally and linguistically diverse, a sustained effort is required to implement interventions by healthcare providers and government agencies. |
| **Health systems/policy** | | | | | |
| (Baron et al., 2016) | Maternal mental health in primary care in five low- | | Prevalence and impact of priority maternal | Situational analysis | No mental health services are dedicated for perinatal women or |

**Table 4.** (*Continued*)

| Year | Title | Location | Focus of study | Type of study | Comments on Policy/systems |
|---|---|---|---|---|---|
| | and middle-income countries: a situational analysis | National and Madhya Pradesh | mental disorders as well as existing policies, plans and services | | strategies in place in PHCs to detect mental health disorders in the Sehore district.<br>Imperative that mental health programmes include information systems to ensure detection and treatment coverage.<br>Awareness of factors that influence access to mental healthcare is important in developing future programmes. |
| (Shrivastava et al., 2015) | Antenatal and postnatal depression: a public health perspective | National | Maternal mental health policy and health systems | Commentary | A universal psychosocial assessment should be undertaken within the primary care system.<br>Health professionals can actively screen the mother and family for stressors. They should also have a high index of suspicion and understand the importance of early detection.<br>Communication skills of physicians should be addressed. |
| (Bagadia and Chandra 2017) | Starting the conversation – Integrating mental health into maternal healthcare in India | National | Need for policies/health systems for maternal mental health | Commentary | Specific recommendations:<br>Upskilling existing community health workers, existing counsellors and psychologists could be identified as points of referral.<br>Sensitising obstetricians and paediatricians to mental health issues to ensure women do not get missed.<br>With the strengthening of the district mental health programme, a mental health specialist in a district or general hospital can be a point of referral.<br>Use of technology to educate health workers. |
| (Ganjekar et al., 2020) | Perinatal mental health around the world: priorities for research and service development in India | National | Maternal mental health policy, health systems and services | Commentary | Too ambitious to have a single plan that works nationwide.<br>A stepped care approach will require education of healthcare professionals.<br>Need to adapt screening tools to the local setting.<br>Need to train healthcare professionals to provide evidence-based interventions and appropriate care pathways. |
| (Priyadarshini et al., 2023) | Recommendations for Maternal Mental Health Policy in India | National | Maternal mental health policy | Commentary | Three policy options proposed<br>1. Strengthening and focused implementation of the National Mental Health Programme<br>2. Integrating mental health in the ongoing Reproductive, Maternal, Newborn, Child and Adolescent Programme<br>3. Including a maternal component in the NMHP<br> Pilot project before nationwide implementation, including three levels with comprehensive planning to address the needs of both urban and rural populations.<br>Information, education and communication campaigns by |

**Table 4.** (*Continued*)

| Year | Title | Location | Focus of study | Type of study | Comments on Policy/systems |
|---|---|---|---|---|---|
| | | | | | community health workers and community leaders will also be essential.<br>Need to increase Human Resources. |
| (Jatchavala et al., 2023) | Perinatal mental health in India and Thailand: A call for collaboration | National | Current state of perinatal mental health in India and Thailand | Commentary | Some mother and baby units and community-based services for perinatal mental health have been established. These services need to be integrated and scaled up within existing national programmes to make sure they are sustainable and accessible.<br>Capacity building for research and service development.<br>Priority should be given to the scaling up of interventions and PMH should be included in policies, guidelines, curricula and legislation.<br>Immediate need to develop PMH service models and integrate these into maternal and child health services.<br>Development of national guidelines and established services for the PMH issues for transgender women and the LGBTQ+ population.<br>International exchanges/consortia among researchers, policymakers, practitioners and patients will help develop locally appropriate and sustainable. Interventions |
| (Kalra et al., 2024) | National policies and programmes for perinatal mental health in India: a systematic review | National | Maternal mental health programmes and policies | Systematic review | No specific national policy or programme on maternal mental health was identified.<br>Universal access to health and mental health including for women and children was articulated in maternity and mental health policies.<br>National policies as well as evidence based locally tailored programmes are needed across the country.<br>The success of maternity care policies provides a platform to address the mental health needs of women in existing care pathways.<br>Screening by non-expert clinicians can be feasible, and tools need to be adapted. These could be accompanied by financial incentives to screen.<br>Population-based public health initiatives.<br>Policies relating to poverty reduction and eliminating gender-based violence are also important in addressing the female gender related risk.<br>Using available digital platforms like the mother and child tracking system and telehealth to deliver interventions.<br>Need to set up mother and baby units across the country as well as community PMH services. |
| (Handa et al., 2024) | Shedding light on maternal mental health in LMICs: a cornerstone of | National and regional | The importance of addressing maternal mental health within the | Commentary | Integrating maternal mental health into MCH programmes is a necessity.<br>One of the main goals of the NMHP is de-stigmatisation – awareness and |

(*Continued*)

**Table 4.** (*Continued*)

| Year | Title | Location | Focus of study | Type of study | Comments on Policy/systems |
|---|---|---|---|---|---|
| | maternal and child healthcare | | framework of MCH programmes | | education around PMI should be given to mothers.<br>Routine screening should be implemented in India, and screening should be culturally sensitive and integrated into the existing healthcare workflow.<br>Multidisciplinary approach to treatment and collaborative care.<br>Primary care mental health initiatives to close the treatment gap.<br>Capacity building for healthcare practitioners and a role for non-specialist employees.<br>Intersectoral cooperation and health system strategies that emphasise life course-based prevention and treatment. |
| (Chauhan and Chauhan, 2024) | Call for Action: The Obstetrician's Role in Peripartum Mental Health | National | Role of OBGYN in maternal mental health | Commentary | The obstetrician is duty-bound to identify those women at risk and should refer appropriately and educate patients and families.<br>Mental healthcare should be delivered seamlessly from antenatal and postnatal clinics by collaborating with on-site mental health professionals.<br>Resources should be allocated to training all levels of healthcare workers. Capacity building for mental healthcare will allow task shifting/task sharing, thus widening the network of maternal mental healthcare.<br>Community-based interventions and using the Mother and Child Tracking System may prove viable. |
| (Ragesh et al. 2017) | Context and scope of social work interventions in Perinatal mental health settings in India | Bangalore | Role of social workers in the perinatal mental health setting | Commentary | A biopsychosocial framework can be adopted during assessment, including intervention based on principles of social justice and human rights.<br>Social workers need to have a role in perinatal mental health settings, Individual, family, group and community level interventions is required. |
| (Shanbhag et al., 2022) | A perinatal psychiatry service in Bangalore, India: Structure and function | Bangalore | Structure and function of NIMHANS perinatal psychiatry services | Chapter | NIMBUS (NIMHANS maternal behaviour) scale was developed for the assessment of a mother's behaviour towards the infant. This can be used easily without formal training and hence could be used in low-resource settings.<br>Understanding women's belief systems and explanatory models is essential in making services user-friendly.<br>Healthcare providers need to be educated about cultural practices and their meaning so that they can advocate for diverse patient populations.<br>24 h perinatal mental health helpline service is available for women across all states in the country.<br>A range of services has been |

**Table 4.** (*Continued*)

| Year | Title | Location | Focus of study | Type of study | Comments on Policy/systems |
|---|---|---|---|---|---|
| | | | | | developed based on a combination of evidence-based practices worldwide and culturally appropriate adaptations. A stepped care approach by up-skilling community healthcare workers and PHC doctors has been trialled successfully |
| **Knowledge/perceptions** | | | | | |
| (Insan et al., 2022) | Perceptions and attitudes around perinatal mental health in Bangladesh, India and Pakistan: a systematic review of qualitative data | National | Perceptions and attitudes of perinatal women, their families and healthcare providers surrounding PMH | Systematic review | Affective symptoms are much less acceptable to women than physical symptoms of perinatal mental illness. Even healthcare providers perceive it to be inappropriate to ask about emotional or mental health symptoms. Increased training and education is required to increase awareness around PMH. Barriers and enablers to accessing PMH care operate at a number of levels, ranging from sociocultural to structural factors. Women resorting to personal and religious coping strategies, but these are not sustainable – calls for culturally appropriate coping interventions for long-term management. The education sector will be a key actor in the development of interventions addressing many of the attitudes and perceptions identified. Improved education for healthcare providers at all levels is vital for increasing awareness of PMH and better knowledge of the symptoms and management options. Should be trained in friendly and confidential counselling. Psychosocial education aimed at tackling IPV, gender inequalities and stigma within communities could be provided through the government and NGOs. Family planning services are also key to addressing the issue of unplanned pregnancies and can be achieved through maternal, neonatal and child health platforms at the community and household level. The social welfare sector should be integrated to tackle poverty, housing and unemployment. |
| (Ransing et al. 2020) | Perinatal depression-knowledge gap among service providers and service utilisers in India | New Delhi Maharashtra | The knowledge gap among service providers- nurses and medical practitioners, as well as service users- pregnant women | Cross-sectional study | The vast majority of pregnant women are uninformed about perinatal depression. There was a misconception about the aetiology and management of perinatal depression among nurses. This is important information as nursing providers are well placed to provide a large part of the multilevel intervention needed for the integration of perinatal mental health. Training of nurses must go hand in hand with creating |

**Table 4.** (*Continued*)

| Year | Title | Location | Focus of study | Type of study | Comments on Policy/systems |
|---|---|---|---|---|---|
| | | | | | awareness among pregnant women, as there are currently significant misconceptions around perinatal depression. |
| (McCauley et al., 2020) | Good health means being mentally, socially, emotionally and physically fit: women's understanding of health and ill health during and after pregnancy in India and Pakistan: a qualitative study | New Delhi | What women consider health and ill health to be, and women's views on screening for mental illness and social morbidity | Qualitative study | Currently, there is good coverage and uptake of antenatal care, but content needs to be adapted to ensure good quality, comprehensive care that is respectful and integrated, that also delivers mental and social care with agreed country-specific content.<br>Most women feel positively towards being asked about mental health, and report that currently they aren't screened for mental illness. |
| (Malik, 2024) | Ignored sufferers: a phenomenological inquiry into the lived experiences of postpartum depression among men | Meerut, Delhi | Lived experiences of men with PPD | Qualitative, phenomenological | Policy-level changes, such as paternal leave for fathers, are needed<br>Need for the provision of medical support for young children through medical coverage and financial support.<br>Encourage open communication among family members to address the issues of men's silence.<br>Need to implement more widespread screening of new fathers for PPD, especially if their partners are experiencing PPD.<br>Counsellors should be aware of factors influencing PPD in men (men's silence in relationships due to fear of conflict, economic burden, sleep deprivation, interference from wider family) and collaborate with other stakeholders to provide holistic care and support.<br>Need for structural guidelines to aid with diagnosis and treatment in men. |
| (Banerjee et al., 2022) | It seemed like my fault for wanting to become a mother – experiences and perceptions related to motherhood in women with severe mental illness | Bangalore | First-person accounts of mothers as they move from pre-conception to pregnancy | Qualitative study with a 'social constructivist paradigm' | Emergent themes were thoughts/feelings about childbearing (including ambivalence, guilt), impact of mental illness (stigma, self-care, effects of medication on foetus), unmet needs (lack of emotional support, doubts on impact on pregnancy) and caregivers' reactions (discrimination, anger/abuse, selective support).<br>Centrality of motherhood and the dual role of patient and mother. These results provide critical insights for service and policy provisions. Being sensitive to these nuances, perceptions will help tailor perinatal care and shape interventions. |
| (Shanbhag et al., 2023) | 'If They Don't Ask, We Don't Share' – A Qualitative Study on Barriers and Facilitators to Discussing Mental Health with Obstetric Care Providers in Urban Anganwadi among Pregnant Women in India | Bangalore | Barriers and facilitators to discussing mental health among pregnant women | Qualitative | To enhance screening and support for PMI, it is necessary to increase awareness about the importance of mental health in the perinatal period among women and families. Education will reduce stigma and increase their ability to have open discussions with healthcare providers. Technology can play an important role- messages regarding |

**Table 4.** (*Continued*)

| Year | Title | Location | Focus of study | Type of study | Comments on Policy/systems |
|------|-------|----------|----------------|---------------|----------------------------|
| | | | | | mental health could be added to already existing voice messages about aspects of regular antenatal care.<br>Train obstetric care providers in sensitive screening and ensure privacy in clinics. The first step in providing mental health support in antenatal settings is facilitating a non-stigmatising, sensitive and private discussion. Train first responders like community health workers in a conversational approach. Collaborative approach and streamlined referral system will ensure women get timely help, and these barriers need to be addressed before starting a screening programme. |
| (Fellmeth et al., 2023) | Women's awareness of perinatal mental health conditions and the acceptability of being asked about mental health in two regions in India: a qualitative study | Kangra, Bangalore | Awareness of PMI and acceptability of being asked about mental health and assessment tools | Qualitative | Existing assessment tools may need translation and cultural adaptation. Asking about symptoms of PMI during routine antenatal and postnatal appointments can help identify women at risk.<br>Being asked about mental health was generally considered to be acceptable, but questions related to suicidality may be challenging in a community setting, requiring sensitivity by the interviewer. Discussions need to occur in a private space, with advanced warning and a supportive and familiar health professional delivering the questions. |
| (Pasricha et al., 2021) | Sense of Coherence, Social Support, Maternal–foetal attachment and Antenatal Mental Health: a Survey of Expecting Mothers in Urban India | Sonipat | Associations of expecting mothers' sense of coherence, perceived social support and maternal-foetal attachment | Cross-sectional | Importance of training OBGYN physicians and nurses performing perinatal care in urban India. Importance of utilising techniques to increase perceptions of social support among pregnant women – for example, healthcare facilities could create support groups for pregnant women.<br>Importance of developing interventions to encourage maternal foetal attachment among urban Indian women.<br>Lamaze classes exist in urban India, which can be modified to include awareness on perinatal mental health, maternal-foetal attachment and what to expect when the baby is born. |
| (Lodha et al., 2022) | Perceptions of perinatal depression among low-income mothers and families in Mumbai, India | Mumbai | Perceptions of perinatal depression among mothers and their relatives | Mixed Methods | Interventions should factor in the target population's awareness levels and sociocultural perceptions. Clinicians must be trained to increase their awareness and sensitivity, and, importantly, to carry out routine screening for perinatal depression.<br>Mental health treatment and family care must be made accessible within maternal health facilities/<br>A multidisciplinary approach is key to battling perinatal depression and PHC; perinatal services and |

**Table 4.**  (*Continued*)

| Year | Title | Location | Focus of study | Type of study | Comments on Policy/systems |
|---|---|---|---|---|---|
| | | | | | community health worker services can be integrated to provide holistic care.<br>Future resource allocation and local/institutional policies should reflect these priorities. |
| **Screening tools** | | | | | |
| (Fellmeth et al., 2021) | Validated screening tools to identify common mental disorders in perinatal and postpartum women in India: a systematic review and meta-analysis | National | Validity of screening tools for perinatal mental illness in India | Systematic review and meta-analysis | Early detection and treatment of perinatal Common Mental disorders (CMD) is essential to minimise the adverse effects and improve outcomes. Current gaps in evidence, specifically around the validity of screening tools for anxiety, hinder the practice of screening.<br>Existing evidence is narrow in scope and focuses entirely on perinatal depression and almost exclusively on the Edinburgh Postnatal Depression Scale (EPDS).<br>Testing the acceptability of screening tools to the local population is essential in informing policymakers.<br>Comparisons of EPDS with other depression screening tools are necessary to establish which is most effective in the Indian setting and guide policymakers and practitioners. |
| (Joshi et al., 2020) | Validation of the Hindi version of the Edinburgh postnatal depression scale as a screening tool for antenatal depression | Madhya Pradesh | Validates a linguistically and contextually appropriate Hindi version of EDPS | Cross-sectional study | The Hindi version of EDPS can be used as a valid measure to screen antenatal depression.<br>Screening for antenatal depression should be included as an essential component of routine antenatal care. Screening of high-risk populations by primary health care workers.<br>There is a specific need for introducing a validated screening instrument and appropriate management tools for PMI as a specific service delivery component in primary health care.<br>Given the ease of administration of EDPS, it can be a beneficial assessment tool, but it will only be meaningful if paired with confirmatory diagnosis and treatment by a psychiatrist. |
| (Ransing et al. 2020) | Assessing antenatal depression in primary care with the PHQ–2 and PHQ–9: Can it be carried out by an auxiliary nurse midwife (ANM)? | Maharashtra | Analyse the inter-rater reliability of PHQ–22 and PHQ–9 between ANMs and clinical psychologists | Prospective, observational | These two scales can be used to screen and assess the severity of AD by either qualified or minimally trained community health workers. The stepped care approach using existing manpower of the RCH programme and nurse-based collaborative-integrative models is an option for integration of mental healthcare into the RCH programme. This will require ANMs to have skills and training, and if good inter-rater reliability is achieved, then there can be successful integration of ANM-based models under the NMHP in India.<br>ANM-administered PHQ–9 could be a reliable method to detect suicidal |

(*Continued*)

**Table 4.** (*Continued*)

| Year | Title | Location | Focus of study | Type of study | Comments on Policy/systems |
|------|-------|----------|----------------|---------------|----------------------------|
| | | | | | ideation in the context of the MCH programme.<br>ANM administration of PHQ–2 and PHQ–9 is a reliable approach to screen antenatal women for depression and rate its severity. The low cost and ease of administration of these scales by ANMs could cover a large number of women across the country.<br>Routine screening by ANM may be considered an appropriate method in a low-resource setting.<br>Need for modification and cross-cultural adaptation of PHQ–9 to improve the performance and reliability of screening depression in rural settings.<br>Will help improve the screening, facilitating early detection, referral to appropriate mental health services and overcome the mental human resource deficit in the future. |
| **Legislation** | | | | | |
| (Aneja and Arora, 2020) | Pregnancy and severe mental illness: Confounding ethical doctrines | National | Mental Health Care Act (MHCA) | Case report | The MHCA currently does not directly address perinatal mental health and some of the unique ethical concerns of mental illness during pregnancy and motherhood.<br>Integration of mental health services with other services like social services, child protection, public health and medico-legal services is important<br>The NIMHANS model for perinatal mental health services may be replicated at some of the well-developed tertiary psychiatric care services. |
| (Behl, 2021) | Perinatal depression during the COVID–19 Pandemic: Need to introduce Perinatal Mental Health Services under the Indian Reproductive Health Rights Framework | National | Impact of COVID–19 on PMH and responses made by India | Narrative review | There have been reports that with adequate training and supervision, ASHA workers and ANMS can detect and screen vulnerable individuals.<br>Integration of mental health services into the maternal healthcare platform is recommended and is evidence-based and cost-effective.<br>Services for high-risk perinatal women, especially those who are facing IPV/DV, should be given priority.<br>Specific medical services, guidelines and information should be made available to vulnerable groups, such as perinatal women, during pandemics.<br>Digital platforms should be used in a culturally acceptable manner, Governments and other stakeholders should act proactively in devising services and guidelines that are digital-friendly |
| (Behl and Nemane, 2023) | Perinatal mental health and the justice delivery system in India | National | How the absence of services for PMI affects the justice system in cases of infanticide | Commentary | A better understanding of these issues by all concerned – the judiciary, advocates, police<br>The Indian justice system needs to urgently adapt to deal with the challenges posed by such complex medico-legal issues in the setting of PMI. Better training is imperative.<br>When a healthcare system does not |

**Table 4.** (*Continued*)

| Year | Title | Location | Focus of study | Type of study | Comments on Policy/systems |
|---|---|---|---|---|---|
| | | | | | include a perinatal mental health assessment, it is unfair to blame a woman when her mental health deterioration goes unrecognised and untreated. |
| (Behl, 2023) | Judicial interface with perinatal depression in India: an empirical analysis and thematic review of published judgements | National | Medico-legal milieu of perinatal depression through the lens of judicial interface | Empirical analysis and thematic review | Imperative to introduce policies for preventing and managing postnatal depression (PND) as well as services which will be beneficial to the judiciary system as well.<br>Imperative that Maternal and Child Health and Mental Health frameworks incorporate PMH services.<br>Necessary to train the judiciary and police about the interface between PND and the justice delivery system.<br>Awareness and sensitisation regarding PND among policymakers, health practitioners and the legal body is critical to protect human rights and the rights of those suffering from PND. |
| (Behl et al., 2024) | Indicators of perinatal mental health of the Indian women, and the National Family Health Surveys: a trajectory of obscurity | National | The inclusion of perinatal mental health in the NFHS results | Commentary | Perinatal mental health services and the status of PMI among women in India must be included in the NFHS framework. This data can provide an evidentiary base for future policy making.<br>The existing absence of any data about PMI shows the lack of recognition of PMI as a public health issue.<br>Multisectoral and collaborative approach must be taken towards curbing the prevalence and consequences of PMI, and the Ministry of Health and Family Welfare must play a cardinal role. |
| (Behl et al., 2024) | Perinatal Mental Disorders: The 'Non-Liquet' Facet of Mental Health Legislative Instruments in India | National | The coverage of perinatal mental disorders under legislative instruments | Commentary | Immediate need to highlight perinatal mental disorders through legislative instruments. The MHCA Act should be amended to explicitly include women during the perinatal period.<br>Integration of perinatal mental health services within the maternal healthcare infrastructure is in consonance with existing legislation and PMI and should be undertaken.<br>Policies and services focused on PMH will align with the goals of the NMHP.<br>Increasing understanding about PMI will influence the approach of society towards other mental disorders and lead to de-stigmatisation.<br>Training existing and future healthcare personnel will also help in fulfilling the objectives of the NMHP. |
| (Behl et al., 2025) | Legal interventions for perinatal depression in India: a qualitative study with clinical specialists | National | Efficacy of existing legal interventions in addressing and managing PND | | Neither is PMH covered nor prioritised under the existing mental healthcare frameworks in India.<br>The absence of a nationwide |

(*Continued*)

**Table 4.** (*Continued*)

| Year | Title | Location | Focus of study | Type of study | Comments on Policy/systems |
|---|---|---|---|---|---|
| | having expertise in perinatal mental health | | | | applicable framework results in a knowledge gap.<br>The MHCA should be amended to ensure explicit inclusion of perinatal women.<br>A nationwide applicable policy should be introduced for the universal screening of PND during antenatal and postnatal appointments.<br>PMH services should be prioritised for women undergoing spontaneous abortion or termination of pregnancy.<br>Change in attitude with curriculum change, soft-skill training and ethics and communications regulations for healthcare personnel.<br>Task sharing approach, policy should provide a stepped care model and facilitate the integration of mental health within the Indian Reproductive Health framework.<br>A multisectoral approach where different ministries collaborate and provide awareness about PND.<br>Tackle disease burden with cautious use of digital healthcare delivery platforms.<br>Models of other countries, including Sri-Lanka, Bangladesh and Pakistan, can be used as a guiding force for PMH service models.<br>Legal interventions can play a crucial role in addressing and managing PMI at the population level as well as protecting the 'right to perinatal health' for Indian women. Amending existing laws can create momentum. |
| **Consequences of perinatal mental illness** | | | | | |
| (Babu et al., 2020) | Small for gestational age babies and depressive symptoms of mothers during pregnancy: results from a birth cohort in India | Bangalore | The possibility of any relationship between the EDPS score and the incidence of SGA babies | Prospective cohort study | Need to universally screen women for depression during pregnancy-healthcare workers at the primary care level can conduct screening.<br>Timely and effective screening, diagnostic services and evidence-based antenatal mental health services are needed to reduce the rates of small for gestational age babies.<br>The government can modify and/or incorporate mental health screening within the existing provisions of the national health mission. |
| (Roy et al., 2022) | Impact of perinatal maternal depression on child development | Kolkata | Effect of antenatal and postnatal depression on child development at 12 months | Prospective observational study | Pressing need to include mental health screening in routine antenatal care.<br>Adequate and timely intervention of AD can help ameliorate the burden of PPD and adverse child development attainments. |
| (Jakhar et al., 2023) | Mental Health Impact of COVID–19 Infection on Postpartum Women from Lower and Middle-income Backgrounds in India and its Effects on Early Mother- | New Delhi | Impact of COVID–19 infection and its effects on bonding during the first 8 weeks postpartum | Cross-sectional Observational | Need for specialised mental health services and individualised breastfeeding interventions for this vulnerable population.<br>Beyond the specific fear of COVID–19, other factors like recent loss of income, psychosocial stressors, |

**Table 4.**  (*Continued*)

| Year | Title | Location | Focus of study | Type of study | Comments on Policy/systems |
|---|---|---|---|---|---|
| | Infant Bonding: An Observational Study | | | | pandemic-related restrictions, limited support and uncertainty create multiple stressors that affect mental health. These factors must be considered when interpreting the study and developing strategies to support the mental health of postpartum women during the pandemic.<br>Psychiatry and maternal care facilities must be aware of the need to alleviate stress associated with caring for a newborn in the pandemic and promote mother-infant bonding despite COVID–19 restrictions.<br>Recommend promoting exclusive breastfeeding practices and screening all post-COVID mothers for distress and enhancing perinatal practices. |
| (Mhamane et al., 2024) | Postpartum depression: its association with IYCF practices and effect on child growth indicators in urban slums of Mumbai, India | Mumbai | Association between PPD and IYCF practices and the effect on child growth during the first 1,000 days of life | Cross-sectional | It is essential to have mental health inclusive policies and programmes for mother and child health, as well as engaging in capacity building of grassroots workers to spread awareness about PPD.<br>Include maternal depression in maternal and child health programmes to ensure the holistic health of mother and child.<br>Spread awareness about the early signs, mandatory screening, providing essential counselling, building mental health inclusive policies, including a psychologist within teams, and engaging in capacity building will help tackle the significant burden of PPD. |

**Figure 3.** Policy and system recommendations word cloud drawn from this review.

Kedare et al., 2024; Mhamane et al., 2024; Easwaran et al., 2025). Routine screening for mental health disorders in antenatal care is widely seen as critically important (Kukreja et al., 2012; Prost et al., 2012; Shrivastava et al., 2015; Sheeba et al., 2019; Amipara et al., 2020; Joshi et al., 2020; Doke et al., 2021; Fellmeth et al., 2021, 2023; Ransing et al., 2021; Shiva et al., 2021; Meerambika Mahapatro et al., 2022; Nisarga et al., 2022; Kalra et al., 2024; Kumari and Basu, 2024; Priya et al., 2024; Rajeev et al., 2024) and should include screening for risk factors for suicide (Supraja et al., 2016), so that at-risk individuals can be prioritised for early intervention (Mariam and Srinivasan, 2009; Shrivastava et al., 2015; Rathod et al., 2018; Amipara et al., 2020; Goyal et al., 2020; Fellmeth et al., 2021; Jelly et al., 2021; Mahale et al., 2021; Kumari and Basu, 2024; Rajeev et al., 2024). Some valid screening instruments exist (Fellmeth et al., 2021), such as EPDS (Joshi et al., 2020); however, these may have limitations when used in 'Non-Western settings'(Shrestha et al., 2016). A number of treatment interventions have been the subject of clinical trials, including the 'Thinking Healthy Programme'(Fuhr et al., 2019; Singla et al., 2021), participation in women's groups (Tripathy et al., 2010) and individual (Prabhu et al., 2025) and group interventions (George et al., 2020) with preliminary evidence of efficacy.

### Perinatal mental health and the community

Through informal interviews with psychiatrists, psychologists and psychiatric nurses, we have gained an understanding of some community beliefs surrounding mental health. The following points were discussed with interviewees:

1. In Telangana, and indeed throughout India, it is common practice that women go back to their family home around the seventh month of their pregnancy and stay until around 40 days after birth, particularly for the birth of the first child. Here, they are well supported, and the practice of isolating the mother during this time seems to be rare. However, when women move back to their husband's family, they often receive less support and are expected to do the housework in addition to looking after the baby. Little is known about how this move may affect the risk of mental health conditions, and this could be explored further in focus groups.
2. Stigma is a major barrier for women in need of mental health support. In families, some level of mental distress is frequently seen to be normal; family members often seem to reject the idea that PMHCs require treatment, help coming from outside is often not accepted and medical interventions can be seen as 'unnecessary'. Consulting psychiatrists and psychotherapists, therapy and medication are widely considered as taboo. Counselling seems to be better accepted but is usually only available through the private system.
3. Male child preference and dowry practices were also mentioned during interviews as factors that might influence a woman's mental health following the birth of a girl child. If a woman delivers two or three baby girls, they may have more anxiety and depression symptoms in their subsequent deliveries.

### Monitoring and evaluation

No monitoring and evaluation of perinatal mental health conditions has or is currently being conducted in the two districts. The 2015/2016 National Mental Health Survey (NMHS) found that states were at different stages of implementing their health management information systems. Only five states (including Chhattisgarh, Gujarat, Madhya Pradesh and Punjab) routinely recorded mental health in their information systems (National Institute of Mental Health and Neurosciences 2017). According to NMHS, 10% of the Indian population suffers from depression and anxiety, and 20% of these are pregnant women or new mothers (National Institute of Mental Health and Neurosciences 2017); however, this survey did not include Haryana or Telangana in its sampling. One study based in rural Haryana estimated the prevalence of common mental disorders in pregnancy to be 15.3% (Jha et al., 2021) with 2.8% affected by major depression and 15.1% affected by anxiety. No comparable study looking at the prevalence of PMHCs in Telangana was found. However, prevalence rates for antenatal depression in South India range between 16.3% (George et al., 2016) and 36.5% (Hegde et al., 2012). Estimates for postnatal depression prevalence in South India range from 19.8% (Chandran et al., 2002) to 34% (Chainani, 2021) and 21.5% (Saldanha et al., 2014) and 29% (Basu et al., 2021) in North India.

### Discussion

This situational analysis evaluated the perinatal mental health context, policies and plans and services in India as well as locally in Telangana and Haryana. There is no specific national perinatal mental health policy or plan. No specific PMH services are available in either Telangana and Haryana. General mental health services for women who experience PMHCs exist in these states and within Siddipet and Faridabad.

The demographic differences between Haryana and Telangana may affect the prevalence of mental illness during pregnancy and may pose different challenges in establishing perinatal mental health provision.

Adolescent pregnancy is an acknowledged risk factor for PMHCs (WHO, 2008), and Adolescent Girls and Young Women (AGYW) may be subject to a number of other intersecting vulnerabilities, such as unintended pregnancy and exclusion from education (Palfreyman and Gazeley, 2022). Rates of adolescent pregnancy are lower in Siddipet (1.0% of 15–19 year olds) than in Faridabad (2.8% of 15–19 year olds). One study based in South India found increased odds of postpartum depression in women aged under 25 years (Doke et al., 2021). This population will be an important target for early intervention as an 'at-risk' group. Mental health services must be integrated into adolescent sexual and reproductive health services, as the Reproductive, Maternal, Newborn, Child and Adolescent (RMNCH+A) strategy advocates. This strategy addresses the major causes of mortality among women and children, aiming to ensure a 'continuum of care' for integrated service delivery across the life course.

Intimate partner violence (IPV) is strongly linked to PMHCs. A systematic review and meta-analysis looking at studies based in both HMICs and LMICs found that 88% of studies established an association between IPV and perinatal depression (Ankerstjerne et al., 2022). Data presented from the NFHS is likely an underestimate of the true prevalence of violence against women (Shah et al., 2021). Indeed, IPV remains one of the few indicators of risk of PMHCs measured in the NFHS, which Behl (2024)) argues needs to be expanded. Despite a lower prevalence of reported intimate partner violence in the NFHS, the rate of reported rape in Haryana is the fourth highest in the country (National Crime Records Bureau, 2022). Haryana has historically had one of the worst sex ratios in the country, a patriarchal culture and a high rate of violence against women (National

Crime Records Bureau, 2022; Parihar et al. 2015). Pressure to have a male child as well as the gender of the newborn has been found to have an impact on the risk for PMHCs (Shidhaye et al., 2017; Kar et al., 2024; Kumari and Basu, 2024) although this is not uniform across the country (Doke et al., 2021). Research is needed in Haryana, in particular, into what state-specific risk factors exist. Women are affected by high rates of intimate partner violence in both states, which will be a key area to target in reducing the burden of PMHCs.

Poor sanitation is linked to adverse pregnancy outcomes (Padhi et al., 2015), and complications during pregnancy can negatively affect women's mental health (Badiya et al., 2020). A systematic review and meta-analysis by Yang et al. found both lower educational level and poor economic status of families to be risk factors for perinatal depression across 31 studies (Yang et al., 2022). The percentage of homes with sanitation and clean water is similar across both districts. Faridabad is a majority urban district, whereas 86% of the population is rural in Siddipet. This may have an impact on support systems (Kishore et al., 2018) as well as accessibility of services (Kar et al., 2024) and coverage of technology. Rural residence has also been found to be a risk factor for perinatal depression (Easwaran et al., 2025), and there may be lower levels of awareness about PMHCs in this population. A lack of familial and social support has been associated with depression during pregnancy (Basu et al., 2021; Pasricha et al., 2021). Policies aimed at improving the socio-economic status of women should be considered an integral part of any perinatal mental health strategy. Multisectoral approaches that tackle poverty, social protection, violence prevention, education and gender disadvantage are called for (Scott et al., 2020; Insan et al., 2022; Behl, 2024; Handa et al., 2024). Some of these are detailed in Figure 2. In 2015, the government launched the Beti Bachao, Beti Padhao programme with the objective of saving and educating the girl child, which has been associated with significant improvements in the sex ratio at birth in Haryana (Gupta et al., 2018). However, the success of other programmes is not so clear. Despite the passing of the Dowry Prohibition Act, this is still widely practiced across India, and one study based in Maharashtra found more than a third of pregnant women admitted that a dowry was exchanged at the time of marriage (Shidhaye et al., 2017). Unsatisfactory reaction to the dowry and a difficult relationship with the in-laws were significantly related to antenatal depression (Shidhaye et al., 2017).

Maternal and infant mortality is lower in Telangana than in Haryana. Indeed, the MMR in Telangana is less than half that in Haryana, and worryingly, the MMR in Haryana has risen to 110 in 2018–2020 from 91 in 2016–2018. Goli et al. found a clustering of high MMR in North-eastern and central regions (Goli et al., 2022). The strongest correlates with MMR were found to be postnatal care, maternal age and nutrition and poor economic status (Goli et al., 2022). In addition, health infrastructure, fertility levels, sex ratio at birth and years of schooling were significantly related to MMR. Indeed, many of these are risk factors for PMHCs as well, and targeting both nutrition and mental illness together is a strategy being implemented currently in Telangana (UNICEF, 2023). The leading causes of maternal deaths are haemorrhage, infection and hypertension (Meh et al., 2022). However, suicide is also a leading (Oates, 2003) and often an underestimated cause of maternal death. There is a need to understand the factors influencing MMR across these states, and targeting these to reduce maternal mortality in Haryana in particular is crucial; efforts can go hand in hand with strategies aimed at reducing the burden of PMHCs.

A high proportion of women attend antenatal and postnatal assessments in both states. However, at around 60%, attendance at antenatal clinics (ANC) is well below the 90% target set by the EPMM Initiative. Improving this is important as antenatal clinic appointments provide the opportunity for mental health intervention as part of integrated care. There are also concerns that the focus on at least four ANC visits distracts from the content and quality of care. One study based in Telangana found low coverage of gestational diabetes and syphilis testing, substantial deficiencies in symptom checking and communication, and only one woman was asked about her mental health (Radovich et al., 2022). The WHO has released new guidelines recommending an increase from four to eight or more ANC visits and emphasising the importance of person-centred care and well-being (WHO, 2016). Particularly in LMICs, pregnancy is a time of high healthcare attendance in a woman's life. It is well established that integrating mental health into routine maternal care is an important strategy. The WHO (WHO, 2022) recommends this strategy and has published guidance for the integration of perinatal mental health into maternal and child services.

Location of perinatal care is another important factor to consider. The institutional birth rate is higher in Siddipet than in Faridabad, with 47.3% of institutional births conducted in a public facility in Siddipet compared with 42.9% in Faridabad (NFHS-5, 2022). Private hospitals are favoured for care (Kamath et al., 2024). Despite initiatives like the national health insurance scheme (Rashtriya Swasthya Bima Yojana RSBY) and then later the Prime Minister's people's insurance scheme and Pradhan Mantri Jan Arogya Yojana (PMJAY) (Sharma and Nambiar, 2024), India continues to exhibit one of the highest rates of Out-of-Pocket Expenditure (OOPE) worldwide (National Health Accounts, 2022). Out-of-pocket expenditure reflects the household's financial burden for healthcare services. Southern Indian States have higher levels of development and OOPE (Sharma and Nambiar, 2024), and rural areas also often have higher rates and burden of OOPE relative to household expenditure (Vasudevan et al., 2019). Consistent with this, average OOPE per delivery is higher in Siddipet than in Faridabad (NFHS-5, 2021). In addition, women suffering from PMHCs may be driven to seek help from private hospitals due to stigma and often seek help when symptoms are more severe. These factors may also increase the financial burden of PMHCs faced by the individual or family.

Another important difference between the two districts is the percentage of deliveries by Caesarean Section (CS). The CS rate in Siddipet is roughly triple that of Faridabad (NFHS-5, 2022). Delivery by CS has been found to be a risk factor for the development of postpartum depression (Doke et al., 2021). It is also associated with increased OOPE (Mishra and Mohanty, 2019), and financial stress can be a significant burden on the family, and is reported to contribute to postpartum depression, particularly in men (Malik, 2024). Expanding health insurance and improving the public healthcare infrastructure are important, particularly in rural areas (Kamath et al., 2024). Policy interventions, including perinatal mental health interventions, must provide cost-contained strategies and ensure equitable utilisation (Kamath et al., 2024).

The NMHP does not include a national perinatal strategy, and the National Mental Health Policy does not include a focus on maternal health. In addition, we found no PMH strategy included in any maternal or child health programmes. There is no formal perinatal mental health plan in either state. On a state and district level, patients can access mental healthcare at district hospitals during the perinatal period. In both Faridabad and Haryana, there

are specialist mental health professionals; however, no specialist perinatal mental health services exist. There is an institute for mental health serving both states; however, these tertiary institutions may have varying capacity to develop specific perinatal mental health services such as mother and baby units. Travel to seek medical attention from more remote communities may be challenging as well as expensive. In addition, the stigma attached to seeking help for a mental illness prevents many women from receiving the care they need. This emphasises the need for care in the community.

A stepped-care model of mental health intervention is recommended by the WHO (WHO, 2022) and makes sense in a predominantly rural setting with relatively few specialist mental health service hubs (Bagadia and Chandra, 2017; Ganjekar et al., 2020; Ransing et al., 2021; Kukreti et al., 2022). This is the most efficient (Ho et al., 2016) way of delivering care as it involves delivering low-resource interventions to the majority of people while retaining the ability to provide more intensive treatment to those who need it. The stepped care package should include perinatal mental health promotion, prevention measures, identification and treatment. Once adequately trained, existing community health workers such as ASHAs and ANMs could have an important role in referral (Ganjekar et al., 2020; Ransing et al., 2020a; Shanbhag et al., 2022), with mental health specialists in district/general hospitals acting as points of specialist intervention (Bagadia and Chandra, 2017). An example of a low-intensity intervention is the Thinking Healthy programme, a WHO-supported programme which can be effectively delivered by community health workers or peers (Fuhr et al., 2019).

Delivering mental healthcare within maternal health services carries with it significant challenges. The most obvious challenge lies in training maternity care workers (Supraja et al., 2016), among whom there is evidence of a knowledge gap concerning perinatal mental health (Ransing et al., 2020b). Mental health teams should have an outreach role in planning and delivering education and training, something which Shidhaye et al. note may require a 'fundamental paradigm shift' in the role of psychiatrists and mental health workers (Shidhaye et al., 2016). Training programmes need to be developed that are sensitive to well-documented cultural factors, and prevailing perceptions and attitudes about mental health (Goyal et al., 2020; McCauley et al., 2020; Chainani, 2021; Shiva et al., 2021; Insan et al., 2022). Virtual training courses have been implemented successfully (Shiva et al., 2021). Training was provided to PHC doctors and MCH supervisors by NIMHANS in Siddipet in October 2023. In Faridabad, mental health awareness raising sessions are run for ANMs, staff nurses and medical officers, as well as annual training on PMH for ASHAs were reported during informal interviews. Developing a sensitive workforce and a safe, non-judgemental environment is important in ensuring women feel positively about available services.

A stepped care model utilising existing antenatal staff has been trialled successfully (Kukreti et al., 2022). The Brief Psychological Intervention for Perinatal Depression (BIND-P) model is not only an intervention but also a pathway for stepping up care with universal screening in MCH settings, risk stratification and appropriate onward referral or brief intervention within the obstetric setting (Kukreti et al., 2022). There are appropriate psychiatric services in both districts, and if more specialist care is needed, larger state-level tertiary psychiatric care can be sought. Liaison between different specialties is an important part of care for PMHCs (Shanbhag et al., 2022). Women are supportive of being asked about their mental health specifically and of obstetric staff

facilitating psychiatric referrals (Shanbhag et al., 2023). A collaborative approach and streamlined referral systems between antenatal care and mental healthcare should be reinforced (Shanbhag et al., 2023).

A recent systematic review also found no specific policy or programme on maternal mental health and calls for a comprehensive approach to policy development (Kalra et al., 2024). Kalra et al. advocate for a range of strategies, including addressing female gender related risks such as inequity and socio-economic disadvantage, as well as addressing risk factors for PMHCs (Kalra et al., 2024). Due to the far-reaching public health implications, incorporation of perinatal mental health into maternity and mental health programmes is of high priority, and the development of perinatal mental health policy can in turn shape evidence-based and locally tailored initiatives (Kalra et al., 2024).

In the absence of a perinatal mental health policy, there have been several recommendations as to how one could be introduced. Priyadarshini et al. proposed three options: (1) strengthening and focused implementation of the NMHP, (2) integrating mental health in the RMNCH+A programme and (3) including a 'maternal' component in the NMHP (Priyadarshini et al., 2023). Our review of antenatal clinic attendance suggests that in both states, although particularly in Telangana, there are already many opportunities for screening of mental illness throughout prenatal care. This literature review found that numerous authors advocated for the integration of mental health into routine maternal care. If this is to occur, inclusion of mental health in the RMNCH+A programme is essential, as the maternal health workforce will need to be up-skilled. Developing a perinatal mental health strategy will involve cross-sector participation and a perinatal mental health focus in both existing mental and maternal/child health programmes.

## Strengths and limitations

This is the first integrated review, to our knowledge, of perinatal mental health to investigate maternal health and socio-economic context as well as systems and policies on a national, state and district level. This broad approach is helpful in understanding the wide range of factors that may influence the success of an intervention. It also systematically scopes policy and system recommendations that have been published in India. We acknowledge some limitations, including the collection of secondary data via desktop search, which may lead to incomplete or outdated data. Once major sites such as the national health mission and the ministry for family health and welfare (both national and state sites) were identified, these were reviewed for any relevant policies and programmes. In addition, by interviewing experts in the field on both a national and district level, major PMH policies and programmes should have been identified. A limitation of the literature search was that only one author screened the publications for relevance. The inclusion criteria were based on a subjective analysis of whether the publication made specific policy/health system/legislation recommendations. There is, therefore, a possibility that some papers were missed by human error. However, this scoping review captures an overview of existing publications and policy/health system themes successfully. An informal or conversational approach was taken to interviewing clinicians, which means that the information gleaned is more limited. But this approach has provided a good introduction to service availability and community beliefs.

## Conclusion

This systematic review found very little evidence for existing perinatal mental health systems, policies, and services in the two districts in Haryana and Telangana. Across India, the evidence gap persists on prevalence, coverage and interventions for PMHCs. In order to adequately support women during pregnancy and after childbirth, it is important to understand their needs and socio-cultural context, and to gain evidence on what interventions are acceptable and effective. National and state-level perinatal mental healthcare plans and policies will be critical to provide guidance and support district managers, researchers and communities in implementing perinatal mental health support and to establish services to bridge the existing PMH treatment gap. The findings from this situational analysis will inform the development of a future intervention for PMH in rural communities in Telangana and Haryana.

**Open peer review.** To view the open peer review materials for this article, please visit http://doi.org/10.1017/gmh.2025.10021.

**Supplementary material.** The supplementary material for this article can be found at http://doi.org/10.1017/gmh.2025.10021.

**Data availability statement.** Data can be accessed through the provided links.

**Acknowledgements.** We thank all individuals who have shared information with us.

**Author contribution.** NV designed the study and oversaw data collection, analysis and write-up. LM led and conducted the review. LM, SR and SY collected the data. LM wrote and revised the article draft. SR, SY, DP, PM, JH and NV contributed to editing the draft for publication. All authors contributed to the article and approved the submitted version.

**Financial support.** This work has been funded by an MRC PHIND grant. NV is funded by the Medical Research Council (Grant MR/Y503319/1), the Oxford MSD Improving Equitable Access to Healthcare grant and is partially supported by the UK Medical Research Council (UKRI) for the Indigo Partnership (MR/R023697/1) award. JH is funded by a UKRI Future Leaders Fellowship. The funder has no role in study design; collection, management, analysis and interpretation of data; writing of the report; and deciding to submit the information for publication, including whether they will have ultimate authority over any of these activities.

**Competing interests.** The authors declare none.

**Ethics statement.** This study has approvals from the University of Oxford Tropical Research Ethics Committee (OXTREC reference number 539–22) and the George Institute for Global Health India Institutional Ethics Committee (IEC; reference: 09/2022).

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
