## [Reviewer Report]

Review comments

This article “Perinatal mental health in India in the states of Haryana and Telangana: A district level situational analysis” is a timely attempt to describe perinatal services in Telengana and Haryana with special reference to two districts within the states. Given the burden and morbidity attached to perinatal disorders and the apparent dearth of pernatal mental health services, the article has high degrees of relevance. It has employed a recommended technique with modifications to this end.

Major Comments

1. The article provides information regarding the country, states and data regarding the districts. A more systematic account of country information, state information and district data , where available ,can be provided. Table 2 –which describes district information needs to refer to its sources and provide an account of perinatal obstetric services too. A comparison of socioeconomic indicators of the districts vis a vis the state , if available , would add to the analysis.

2. The discussion section has brought in aetiological factors which were not mentioned in the results section. The implications of these factors may be discussed instead, with the evidence for apparent aetiological factors going to the results section.

3. The discussion section needs to dwell on not only the dearth of perinatal mental health policies, legislation and services , but also the limitations due to the quality of evidence.

4. “Informal interviews” were employed. Details regarding who was interviewed and the interviewers’ where they were from and the reasons why a more formal qualitative process was not employed during the process can be described.

Minor comments

1. Lines 11-13: “One study in the UK estimated the long-term economic and social cost of perinatal depression, anxiety and psychosis to be around £8.1 13 billion (Bauer et al., 2014) for each one-year cohort of births in the UK”

Comment: Are there any studies dealing with socioeconomic costs of perinatal disorders from LAMICs?

2. Lines 214-217 “National PMH inpatient and outpatient services exist at the National Institute for Mental Health and Neurosciences (NIMHANS) in Bangalore (Perinatal Psychiatry NIMHANS) also home to India’s only mother and baby unit.”

Comment: How does Nimhans provides national services? Though the faculty and institution do drive research and perhaps influence policy.

3. Lines 251-255 “These statements indicate that there is a three tier system developed or in the making with training provided and services being in place through IMH.”

Comment: Please provide references for these.

4. Lines 267-271”Accredited Social Health Activists (ASHAs) are also given a day of annual training on antenatal and postnatal mental health, to allow them to recognise symptoms,

provide basic counselling and identify those that need further help. Basic psychiatric medicine (mostly anti-depressants are available at the PHCs and CHCs). District counselling services exist in Haryana (14th Common Review Mission, 2021).

Comment: Please specify what the source of information is for the above lines.

5. Lines 273-274 ”3.1 Perinatal mental health policies, systems and legislation literature review”

Comment: This section is a review of studies on prevalence and intervention from India , not the areas of focus Haryana or Telengana? Are there any studies from these states?

6. Lines 277-278 “Commentary on policy/systems was drawn from the text although exact wording is not used.”

Comment: line 277-278 can be avoided

7. Lines 324-326 “Through informal interviews with mental health experts including psychiatrists, psychologists and psychiatric nurses we have gained an understanding of some community beliefs surrounding mental health”.

Comment:A more specific mention of the credentials/training levels of the interviewees is warranted , as is the interviewing process.

8.Lines 343-346 “In addition, travel to district hospitals can be challenging and expensive for women and their families. For instance, it can take around two hours from the furthest village in Faridabad to travel to the District Hospital by car, and there are no feasible public transport options”.

Comment: This point is best placed in the section on perinatal mental health services.

9.Lines 349-350 “No monitoring and evaluation of PMH problems has or is currently being conducted in the two districts.”

Comment: Monitoring and evaluation(or the lack of monitoring and evaluation in the districts) has focused on the districts, while other sections have focused on the country and states. Consider rectifying this.Clarify what is being referred to as “PMH problems”.

10. Lines 353-358: “This situational analysis evaluated PMH context, policies and plans and, services in India as well as locally in Telangana and Haryana. There is no specific national PMH policy or plan. National PMH services do exist in NIMHANS, and certain states have trialled a number ofdifferent PMH interventions. No specific public PMH services are available in either Telangana and Haryana. General mental health services for women who experience PMH problems exist in these states and within Siddipet and Faridabad”

Comment: NIMHANS is an institution which may drive policy , but whose clinical services do not necessarily have a national outreach.

11. Lines 361 – 366: “Haryana, has a higher population density, with better sanitation, water and electricity provision than seen nationally, along with higher than average figures for life-expectancy and female literacy, and lower rates of adolescent pregnancy. Telangana has a lower

population density than the national Indian average and better sanitation than the national average but below that found in Haryana.’“

Comment: Please give references for these figures- both Telengana and Andhra.

12. Lines 374-377: “Telangana saw a large decrease in rates of adolescent pregnancy from over 10% in 2015-16 to less than 6% in 2019-2021. Continuing efforts to reduce this even further is important. In addition, mental health services must be integrated into adolescent sexual and reproductive health services, as the RMNCH+A advocates.

Comment: An explanation and reference to the RMNCH A+ may be warranted

13. Lines 399-408-

Comment: Where as there are links between Maternal morbidity rates and Perinatal mental disorders, the information may be made more specific and avoid repetition. This information needs to be shifted to the results section and implications of the results are to be incorporated in the discussion section.

14.Line 449: “Delivering mental health care within maternal health services carries significant challenges”

Comment: Whereas training personnel from MCH services is essential, a description of recommended links to formal mental health services where necessary is warranted.

---

## [Reviewer Report]

I wish to congratulate the authors for this important work on perinatal mental health (PMH) in India.

Following are my suggestions for the authors to consider.

Introduction

Consistency in use of terms such as perinatal mental illness/ disorder/ problems

To consider using India/South Asia/LMICs related literature than evidence from high income countries to support the rationale of the study. E.g., there is recent economic evidence available for risk factors/ socioeconomic costs of perinatal depression in South Asia

A more detail is needed emphasizing interstate diversity (comparison of north & south on maternal health service delivery/care & data) in India especially focusing on Telangana & Haryana in India & their differences & authors’ expectations in results/rationale for their assumptions

Methods

Socioeconomic determinants- explanation for including very few parameters given the availability of information on various indicators, some at state & others at district level in India.

Similar concerns about maternal health indicators

How many experts were interviewed & their speciality/expertise?

Scoping review- any reasons to exclude India specific database like Indmed?

Details of papers excluded for no specific policy or system suggestions- it is difficult to conceive to have a peer reviewed publication without implications

Exploration of any published or unpublished data availability on direct & indirect indicators of PMH in both districts

PMH conditions- again different terms being used throughout the script For perinatal mental morbidity

Results

A brief explanation about use of 2011 census in 2025 analysis will be helpful for the readers.

Mixture of findings based on NFHS-4 & 5, better to use recent data

Once the suggestions in methods are possibly considered, more pertinent results may be added for helpful comparison of both the districts

What other national policies and acts were considered that affect maternal mental health

Uncertain about the need for inclusion of NIMHANS, Bengaluru, Thayi Card, Amma Manasu as already introduced in Introduction – result focussed on Telangana & Haryana might be more relevant & addressing the existing differences in service delivery & sociocultural context

Table 3 – I suggest re-arrangement either on the level of evidence or national/state or burden/interventions or separating into different small section to make it easy for readers to understand the recent research. It is hard to extract the information by arranging according to the year only

PMH & the community

Any other gender & culture-based risks- no information on male child preference, dowry, IPV during pregnancy- important to correlate with baseline differences

Discussion- can be enriched further by considering the following suggestions

Line 355- In my opinion, treating patients from across the country is not a national PMH service. The emphasis need to highlight the lack of PMH services in the country & the focussed districts/states + regions

Burden comparison of perinatal mental disorders of different states in the country, if available or at least regions & its influence with analysed determinants in both the states need to be discussed

Inclusion of district level specific implications for both the districts

Challenges in delivering PMH care with high use of private health care utilization and associated financial costs across the country need to be included.

Lack of communication between MH & Obstetrics sector or general linkages between primary & secondary public & privatecare

More discussion of non-health sector policies like poverty reduction, gender equality, IPV

Is this the first review to investigate national systems & policies for PMH?

---

## [Editor Report]

You will see that 2 experts have reviewed your paper and they have raised significant concerns about some aspects of each section. Nevertheless, I am willing to give you the opportunity to revise your paper and respond to each of the reviewer’s comments.

---

## [Reviewer Report]

Thank you for considering suggestions. I am satisfied that it is ready for acceptance and will be an an important contribution to the literature.